# A timer gene network is spatially regulated by the terminal system in the *Drosophila* embryo

Erik Clark[1,2,3]*[†], Margherita Battistara[1,4], Matthew A Benton[1,5]*[‡]

[1]Department of Zoology, University of Cambridge, Cambridge, United Kingdom; [2]Department of Systems Biology, Harvard Medical School, Boston, United States; [3]Department of Genetics, University of Cambridge, Cambridge, United Kingdom; [4]Department of Physiology, Development and Neuroscience, University of Cambridge, Cambridge, United Kingdom; [5]Developmental Biology Unit, EMBL, Heidelberg, Germany

**Abstract** In insect embryos, anteroposterior patterning is coordinated by the sequential expression of the 'timer' genes *caudal*, *Dichaete*, and *odd-paired*, whose expression dynamics correlate with the mode of segmentation. In *Drosophila*, the timer genes are expressed broadly across much of the blastoderm, which segments simultaneously, but their expression is delayed in a small 'tail' region, just anterior to the hindgut, which segments during germband extension. Specification of the tail and the hindgut depends on the terminal gap gene *tailless*, but beyond this the regulation of the timer genes is poorly understood. We used a combination of multiplexed imaging, mutant analysis, and gene network modelling to resolve the regulation of the timer genes, identifying 11 new regulatory interactions and clarifying the mechanism of posterior terminal patterning. We propose that a dynamic Tailless expression gradient modulates the intrinsic dynamics of a timer gene cross-regulatory module, delineating the tail region and delaying its developmental maturation.

**\*For correspondence:**
ec491@cam.ac.uk (EC);
matthewabenton@gmail.com (MAB)

**Present address:** [†]Department of Genetics, University of Cambridge, Cambridge, United Kingdom; [‡]Developmental Biology Unit, EMBL, Heidelberg, Germany

**Competing interest:** The authors declare that no competing interests exist.

## Editor's evaluation

Through the use of multiplexed in situ hybridisation with careful embryo staging, this article represents exemplary documentation of dynamic gene expression patterns in early fly development. By comparison of these patterns in various mutant combinations, a simple logical model for the specification of expression is proposed. This article will be of broad significance to developmental biologists interested in embryo segmentation and gene regulatory networks underpinning patterning.

## Introduction

Insect segments are patterned by a relatively conserved gene regulatory network, including gap genes, pair-rule genes, and segment-polarity genes (reviewed in *Nasiadka et al., 2002*; *Hughes and Kaufman, 2002*; *Clark et al., 2019*). Within and across species, embryonic development depends on these network components being activated at the right times and in the right places. Locally, the maturation of any given segment involves segmentation genes being activated in a conserved temporal sequence (e.g., primary pair-rule genes before secondary pair-rule genes and segment-polarity genes; *Akam, 1987*; *Baumgartner and Noll, 1990*; *Schroeder et al., 2011*; *Clark and Akam, 2016*). Globally, the relative timing of segmentation across the anteroposterior (AP) axis correlates with the specific developmental mode of each species, ranging from predominantly sequential,

## Box 1. Notes on terminology.

Morphological segments are offset from the initial metameric subdivisions of the embryo, the parasegments, by about 2/3 of a segment repeat (*Martinez-Arias and Lawrence, 1985*; *Lawrence, 1985*; *Ingham et al., 1985*; also see *Figure 1C*). The *n*th parasegment boundary (PSBn) refers to the anterior boundary of parasegment *n*.

Segment-polarity stripes are conventionally numbered according to the parasegment they are located within (*Baker, 1987*; also see *Figure 1A and C*). Thus, the first *en* stripe is en1 because it marks the anterior of parasegment 1, and the 14th *en* stripe is en14. The first *wingless* (*wg*) stripe, expressed just anterior to en1, is wg0, and the 14th wg stripe, expressed just anterior to en14, is wg13.

The term telson has been used to refer to the posterior region of the *Drosophila* embryo/larva (usually everything posterior to A8, sometimes everything posterior to A7; *Lohs-Schardin et al., 1979*; *Sato and Denell, 1986*; *Nüsslein-Volhard et al., 1987*; *Perkins and Perrimon, 1991*). As 'telson' generally refers to a terminal non-segmental region of an animal (*Snodgrass, 1935*), or at least its most posterior segment, it is non-standard to use this word to refer to a region that contains more than one segment. We therefore use the more neutral term 'tail' (*Jürgens, 1987*) to refer to the region posterior to PSB14 and anterior to the hindgut.

germband-based patterning in the cricket *Gryllus bimaculatus* or the beetle *Tribolium castaneum*, to more or less simultaneous, blastoderm-based patterning in the fruit fly *Drosophila melanogaster* (reviewed in *Davis and Patel, 2002*).

Previously, we have proposed that segment patterning is coordinated by an underlying framework of 'timer gene' (alternatively, 'timing factor') expression, which broadly regulates segmentation gene expression in time and space (*Clark and Peel, 2018*; *Clark et al., 2019*). We identified the timer genes (not necessarily exhaustively) as *caudal* (*cad*; *Mlodzik et al., 1985*; *Macdonald and Struhl, 1986*), *Dichaete* (*D*; *Russell et al., 1996*; *Nambu and Nambu, 1996*), and *odd-paired* (*opa*; *Benedyk et al., 1994*), all of which code for transcription factors. The expression dynamics of these genes correlate with the progression of segmentation: in *Drosophila*, they are expressed sequentially within the blastoderm, while in *Tribolium* the same expression sequence occurs in cells emerging from the segment addition zone into the segmented germ band (*Schulz et al., 1998*; *Copf et al., 2004*; *El Sherif et al., 2014*; *Clark and Peel, 2018*). In addition, the protein products of these genes are known to directly regulate many segmentation genes in *Drosophila* (*Rivera-Pomar et al., 1995*; *Schulz and Tautz, 1995*; *La Rosée et al., 1997*; *Häder et al., 1998*; *Ma et al., 1998*; *Clark and Akam, 2016*; *Vincent et al., 2018*; *Soluri et al., 2020*; *Koromila et al., 2020*).

However, we currently do not understand how the timer genes themselves are spatiotemporally regulated within the embryo. What accounts for their local sequential activation in segmenting tissues, and why are these dynamics so deeply conserved across species? How is their expression globally regulated along the AP axis, and why is this regulation so evolutionarily flexible?

Here, we investigate these issues in the *Drosophila* embryo, exploiting the fact that segmentation in this model species is not quite so simultaneous as it is often described. Although most of the *Drosophila* blastoderm is patterned simultaneously before gastrulation, the most posterior part of the segmental ectoderm is not patterned until germband extension (*Kuhn et al., 2000*). This 'tail' region (see *Box 1*) is located posterior to abdominal segment 8 (A8) and anterior to the prospective hindgut, and eventually gives rise to a set of ectodermal structures known as the embryonic terminalia (*Turner and Mahowald, 1979*; *Sato and Denell, 1986*; *Jürgens, 1987*). Consistent with the timer gene hypothesis, the tail exhibits *cad*, *D*, and *opa* expression dynamics which differ from those in the rest of the trunk (*Macdonald and Struhl, 1986*; *Russell et al., 1996*; *Clark and Akam, 2016*; *Clark and Peel, 2018*), correlating with the difference in segmentation dynamics.

The patterning of the tail region is dependent on the posterior terminal system (reviewed in *Perkins and Perrimon, 1991*), and, in particular, on its downstream effector, Tailless (Tll; *Strecker et al., 1986*; *Pignoni et al., 1990*). Tll has well-characterised effects on gap gene expression (*Jaeger,*

*2011*; *Janssens et al., 2013*), but its contribution to timer gene regulation is relatively unexplored. As a consequence, the specific regulatory interactions that mediate tail patterning remain unknown (*Casanova, 1990*; *Wu and Lengyel, 1998*; *Smits and Shvartsman, 2020*).

In this study, we discover that *Drosophila* timer gene expression is shaped by a combination of cross-regulatory interactions and extrinsic spatiotemporal inputs. Using multiplexed hybridisation chain reaction in situ hybridisation (HCR ISH; *Choi et al., 2016*; *Trivedi et al., 2018*; *Choi et al., 2018*), we first show that the tail region gives rise to two sets of parasegment-like boundaries after gastrulation, clarifying its segmental nature. We then characterise timer gene expression in wild-type embryos, timer gene mutants, and terminal system mutants, uncovering 11 new regulatory interactions within the *Drosophila* AP patterning network. Using a simple logical model, we show that the revised network both explains wild-type patterning dynamics and recapitulates the mutant phenotypes we examined. We conclude by discussing which aspects of timer gene regulation are likely to be conserved or divergent across species.

## Results
### Two parasegment-like boundaries form sequentially from the *Drosophila* tail region after gastrulation

The *Drosophila* embryo is well-known for its simultaneous mode of segmentation, in which a segmental pattern is laid down at the end of the blastoderm stage, prior to significant morphogenetic movements. Fourteen prospective parasegment boundaries appear at this stage, marked by segmental stripes of segment-polarity gene expression (*DiNardo et al., 1985*; *Baumgartner et al., 1987*; *Baker, 1988*; *Lee et al., 1992*; *Grossniklaus et al., 1992*).

Sandwiched in between parasegment boundary 14 (PSB14; see *Box 1*) and the broad posterior domain of *wg* (thought to correspond to prospective hindgut; *Baker, 1988*) are about four cell rows of ectoderm that remain unpatterned by segment-polarity genes at the end of the blastoderm stage (*Figure 1A*, stage 6). This 'tail' region (see *Box 1*) goes on to form the most terminal structures of the larva (*Turner and Mahowald, 1979*; *Jürgens, 1987*), including a 15th parasegment boundary (*Kuhn et al., 1995*; *Kuhn et al., 2000*), various sensory organs (*Sato and Denell, 1986*; *Jürgens, 1987*; *Kuhn et al., 1992*), and the anal pads (external organs involved in ion transport; *Jarial, 1987*).

The segmental nature of the tail is unclear. The tissue just posterior to PSB15 is abdominal segment 10 (A10; *Figure 1C*). Some authors consider the region to contain a cryptic 11th abdominal segment as well (*Jürgens, 1987*; *Baumgartner et al., 1987*), but most do not (see Discussion: 'The segmental character of the *Drosophila* tail') and, to the best of our knowledge, a 16th parasegment boundary has not been described. To investigate this issue, we used multiplexed HCR ISH to re-examine the expression of the parasegment boundary markers *wingless* (*wg*; *Baker, 1987*; *Rijsewijk et al., 1987*), *engrailed* (*en*; *Kornberg et al., 1985*; *Fjose et al., 1985*), *sloppy-paired* (*slp*; *Grossniklaus et al., 1992*), and *even-skipped* (*eve*; *Macdonald et al., 1986*) during germband extension and extended germband stages (*Figure 1*; *Figure 1—figure supplement 1*).

#### *wg* and *en* expression in the tail
The *wg* and *en* stripes associated with PSB15 emerge during germband extension (*Figure 1A*, stages 8.3–8.4). In contrast to published descriptions of *wg* expression (*Baker, 1987*; *Baker, 1988*), we identified an additional *wg* stripe, wg15, which appeared after germband extension (*Figure 1A*, stage 11.1). During subsequent development, a medial patch of *en* expression appeared posteriorly adjacent to wg15 (*Figure 1A*, stage 11.2). This 'en16' domain is clearly not a full stripe as found in parasegment boundaries 1–15. However, the domain marks the median neuroblast lineage of abdominal segment 10 (*Birkholz et al., 2013*), and median neuroblasts always originate from posterior segment compartments (*Bate, 1976*; *Doe, 1992*; *Biffar and Stollewerk, 2014*). wg15 and en16 therefore seem to correspond to a vestigial 16th parasegment boundary within the *Drosophila* embryo (*Figure 1C*).

#### *slp* and *eve* expression in the tail
In the simultaneously segmenting region of the embryo (here, termed the 'trunk'), segment-polarity domains are initially patterned by stripes of pair-rule gene expression (*DiNardo and O'Farrell, 1987*; *Jaynes and Fujioka, 2004*; *Clark, 2017*). In the tail, PSB15 is prefigured by pair-rule gene stripes slp14

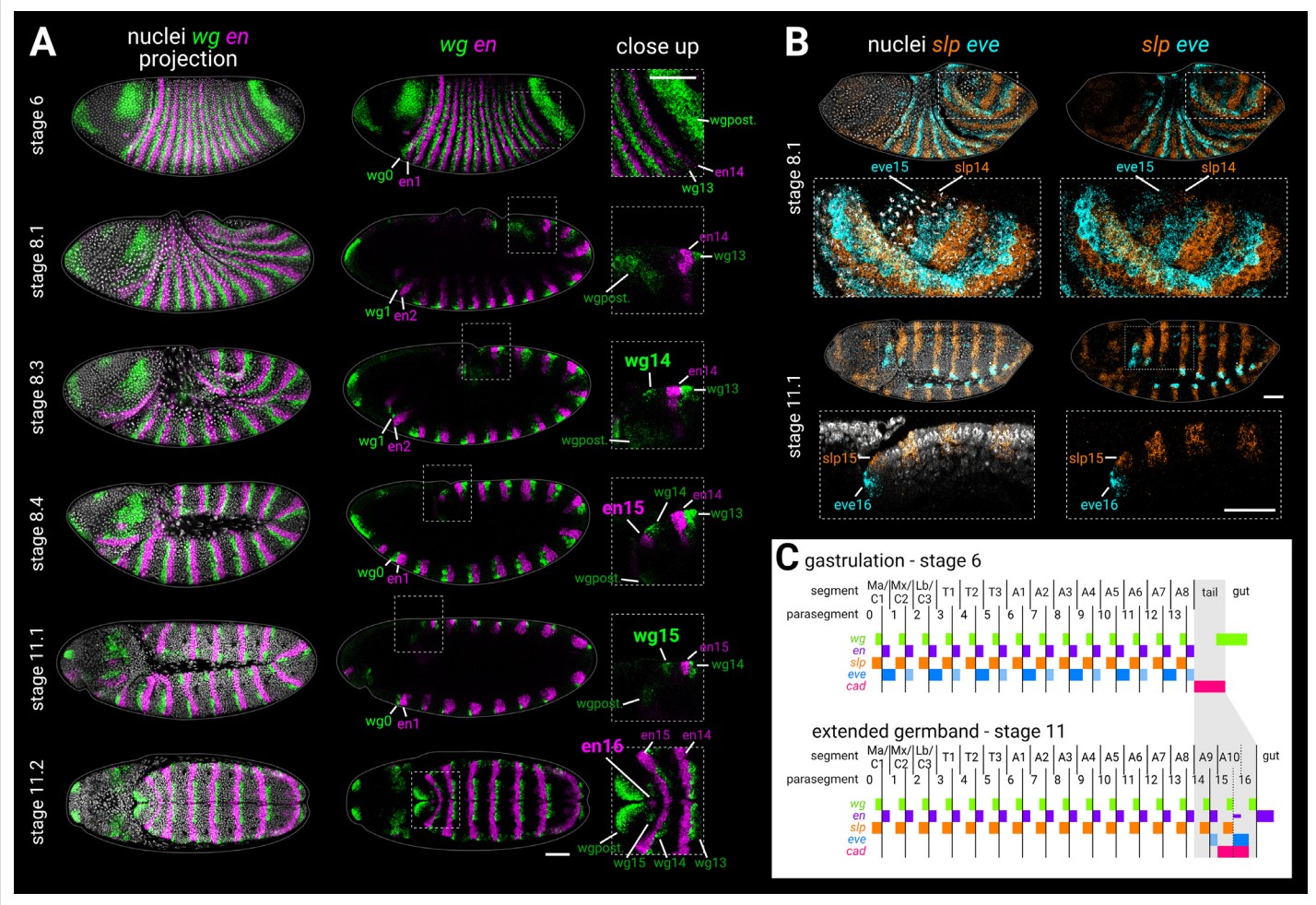

**Figure 1.** Segmentation of the tail region after gastrulation. (**A**) *wg* and *en* expression from gastrulation to extended germband. Left column shows merged maximum projections of *wg*, *en*, and DAPI (nuclei). Middle column shows merged *wg* and *en* expression, either maximum projections (stage 6, stage 11.2), or sagittal sections (stage 8.1 to stage 11.1). Enlarged close-ups of the boxed regions are shown in the right column. Key expression domains are annotated with labels; newly established domains are shown in large font; wgpost = *wg* posterior domain. Stages 6–11.1 show lateral views, stage 11.2 is a 'dorsal' view that actually mainly shows the ventral side of the posterior germband due to germband extension. (**B**) *slp1* (*slp*) and *eve* expression during the division of mitotic domain 4 (stage 8.1) and at extended germband (stage 11.1). Both stages show dorsolateral views. Left column shows a merge with DAPI (nuclei); right column shows gene expression alone. Enlarged close-ups of the boxed regions are shown below the whole embryo views; see Appendix 2: 'Embryo images' for details of how the close-up for stage 11.1 was re-sliced. Key expression domains are annotated with labels. (**C**) Schematic diagram showing the expression of key segmentation genes before tail segmentation (stage 6) and after tail segmentation (stage 11). The tail region is shaded in grey; note the expansion of the region due to morphogenesis, and the refinement of the *cad* domain. PSB16 is shown as a dotted line due to its vestigial nature; en16 is also depicted as narrower than the other domains. Lighter shading for *eve* domains represents weaker or decaying expression. C1-3, gnathal segments; T1-3, thoracic segments; A1-10, abdominal segments; Ma, mandibular segment; Mx, maxillary segment; Lb, labial segment. All embryos are anterior left, dorsal up. Scale bars = 50 µm; grey lines show embryo outlines.

The online version of this article includes the following figure supplement(s) for figure 1:

**Figure supplement 1.** Single-channel images.

and eve15, which appear after gastrulation (*Macdonald et al., 1986*; *Grossniklaus et al., 1992*; Kuhn et al., 2000). We found that slp14 and eve15 emerged simultaneously early in germband extension (*Figure 1B*, stage 8.1), at around the same time as the polarised cell divisions of mitotic domain 4 (*Foe, 1989*; *da Silva and Vincent, 2007*). At the end of germband extension, we were surprised to find that an additional set of abutting *slp* and *eve* stripes, slp15 and eve16, emerged posterior to PSB15 (*Figure 1B*, stage 11.1), in the same region as wg15 and en16. This finding supports our conclusion that wg15 and en16 are segmental in nature.

To the best of our knowledge, the slp15 domain has not been described previously. Persistent *eve* expression at the posterior of the embryo is well-known, although it has been described as a remnant of eve15 (*Macdonald et al., 1986*; *Frasch et al., 1987*; *Sackerson et al., 1999*; *Kuhn et al., 2000*) or the 7th *eve* pair-rule stripe (*Singer et al., 1996*) rather than a separate domain. (Note that eve15 is described by some authors [e.g., *Sackerson et al., 1999*] as the 8th stripe of *eve*, not counting the seven 'minor' *eve* stripes that appear at even-numbered parasegment boundaries just before gastrulation).

## Summary

We propose that two parasegment-like boundaries form sequentially from the tail region of the *Drosophila* embryo after gastrulation (*Figure 1C*). In both cases, segment-polarity gene expression is preceded by a template of abutting *slp* and *eve* expression, similar to the odd-numbered parasegment boundaries of the trunk (*Lawrence et al., 1987*; *Cadigan et al., 1994*). Unlike in the trunk, however, the resolved segmental *eve* stripes appear de novo and are not preceded by a pair-rule phase of expression.

## Timer gene expression differs between the trunk and the tail

Given that *Drosophila* shows distinct segmentation dynamics in the trunk and the tail, we examined the expression of the timer genes (*cad*, *D*, and *opa*) in these regions during blastoderm stages and early germband extension (for an earlier survey using an inferior in situ hybridisation method, see *Clark and Peel, 2018*). To account for the movement of nuclei/cells during blastoderm (*Keränen et al., 2006*) and gastrulation stages, we co-stained the timer genes with *wg* and used the posterior *wg* domain as a fiducial marker. (The posterior *wg* domain appears to be stable relative to nuclei, as nuclear transcription foci are not offset anteriorly or posteriorly relative to cytoplasmic transcripts.) To aid with fine-scale staging of embryos, we have divided stage 5, which lasts ~40 min at 25° C, into five timeclasses based on gene expression and morphology (see Appendix 1).

### Timer gene expression in the trunk

In the trunk, *cad*, *D*, and *opa* transcripts are expressed sequentially over stages 4–6; first *cad*, then *D*, then *opa* (*Figure 2*; *Figure 2—figure supplement 2*). Despite some AP intensity modulation (presumably downstream of gap and pair-rule genes), similar temporal dynamics are present across the whole trunk region, consistent with its simultaneous mode of segmentation. *cad*, which is maternally deposited and then zygotically expressed, clears from the trunk by stage 5.4 (*Levine et al., 1985*; *Mlodzik et al., 1985*; *Hoey et al., 1986*; *Macdonald and Struhl, 1986*; *Mlodzik and Gehring, 1987a*; *Schulz and Tautz, 1995*). *D*, which is detectable from stage 4.1 (nuclear cycle 10), reaches appreciable levels at stage 4.4 (nuclear cycle 13), rapidly reaches a very high peak at stage 5.2, then declines sharply, with residual expression clearing by stage 6, replaced ventrally by persistent expression in the neuroectoderm (*Russell et al., 1996*; *Nambu and Nambu, 1996*). Finally, *opa* appears at stage 5.1, rapidly builds to high levels, then tapers off during germband extension (*Benedyk et al., 1994*; *Clark and Akam, 2016*).

Cad, D, and Opa protein dynamics broadly match their respective transcript dynamics, albeit with time-lags for synthesis and decay (*Figure 2—figure supplement 3*; *Figure 2—figure supplement 4*). Cad levels decrease steadily in the trunk over stage 5 (see Figure 2B in *Surkova et al., 2008*). D levels rise and fall gradually from stage 4.4 to stage 6, peaking at mid stage 5 (*Figure 2—figure supplement 3*; *Figure 2—figure supplement 4C*). Finally, Opa levels increase throughout stage 5 and into stage 6 (*Figure 2—figure supplement 4B*; see also the live quantification of llama-tagged Opa in *Soluri et al., 2020*). Segmentation stages in the trunk are therefore characterised temporally by decreasing Cad levels, increasing Opa levels, and a pulse of D expression in between (*Figure 2—figure supplement 4D*).

### Timer gene expression in the tail

In the tail, a similar *cad/D/opa* expression sequence is evident, but delayed with respect to the trunk (*Figure 2*). *cad* is expressed continuously in the tail region throughout stage 5 and into germband extension. In contrast, *D* and *opa* expression in the tail region remains either low (*D*) or absent (*opa*) through most of stage 5. At stage 5.4, a *D* tail domain emerges within the lateral part of the *cad* tail

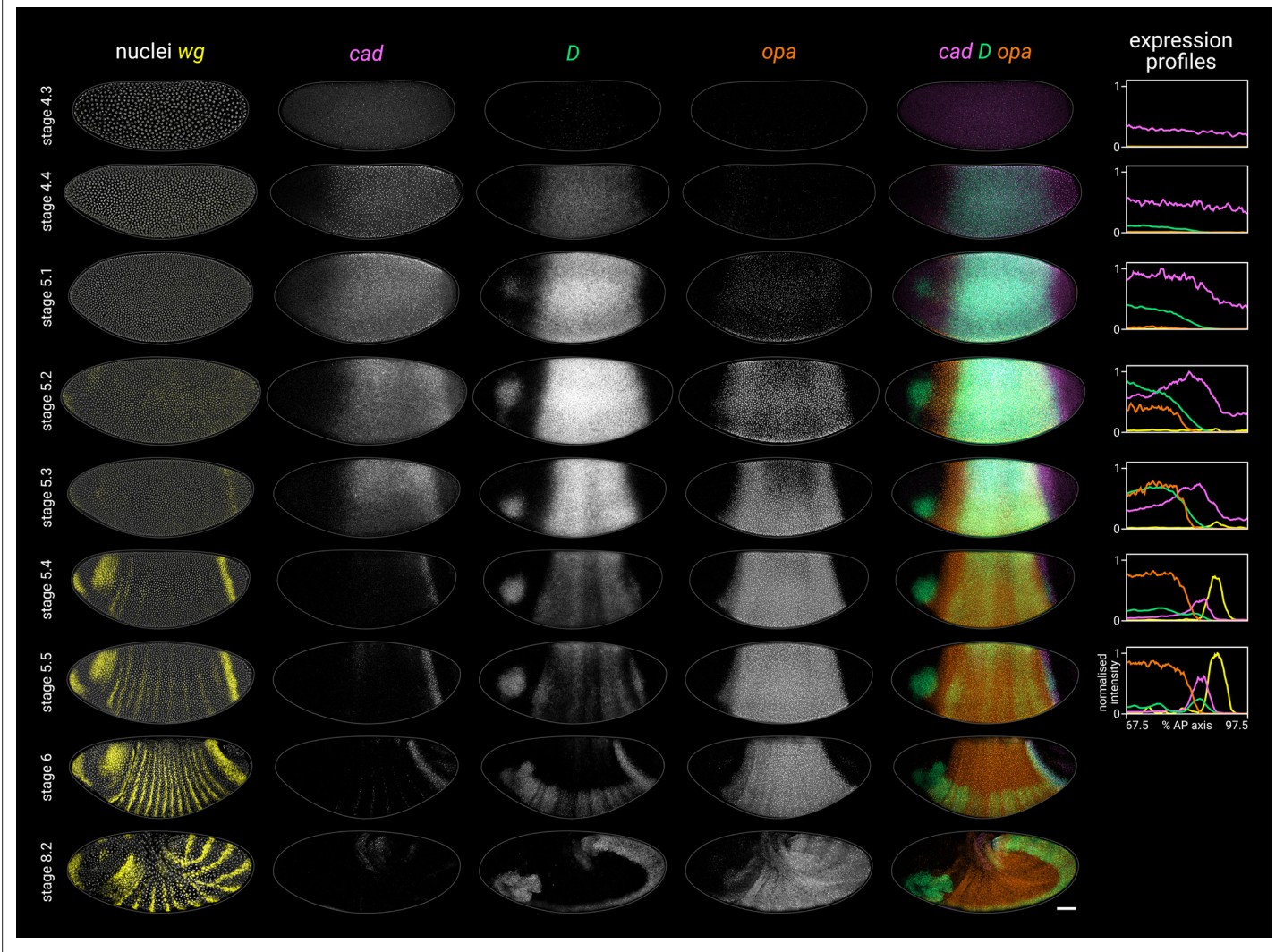

**Figure 2.** Timer gene expression dynamics in wild-type embryos. Column 1 shows a two-channel *wg* and DAPI (nuclei) merge for embryos of gradually increasing age; columns 2–4 show *cad*, *D*, and *opa* channels from the same embryos; column 5 shows a three-channel *cad/D/opa* merge. The plots at the right show quantitative expression traces (67.5–97.5% AP axis; all measurements from the anterior pole) for all four genes, extracted from the embryos pictured to the left. The stage 4.3, stage 4.4, and stage 5.1 embryos are from a different scanning session compared to the rest of the figure. All embryos are anterior left, dorsal up. Stages 4.3–6 show lateral views; stage 8.2 is dorsolateral. Scale bar = 50 μm; grey lines show embryo outlines.

The online version of this article includes the following source data and figure supplement(s) for figure 2:

**Source data 1.** Expression trace source data.

**Figure supplement 1.** Additional single-channel images.

**Figure supplement 2.** Additional early stage embryos.

**Figure supplement 3.** D antibody staining in embryos from stage 4 to stage 6.

**Figure supplement 4.** Expression dynamics of timer gene transcripts and proteins.

**Figure supplement 4—source data 1.** Expression trace source data.

domain, rapidly strengthening and extending dorsoventrally. D protein becomes prominent in the tail domain at stage 6 (*Figure 2—figure supplement 3*; *Figure 2—figure supplement 4C*), again reflecting a modest time-lag for protein synthesis. Finally, *opa* expression expands into the tail region from late stage 5 (described below).

High-resolution close-ups of nascent transcripts, mature transcripts, and synthesised protein (*Figure 3*; *Figure 3—figure supplement 1*) reveal subtle posterior shifts. The *cad* tail domain is mostly anterior to the *wg* posterior domain, with an overlap of a single cell row (*Figure 3A*, *cad/wg* merge).

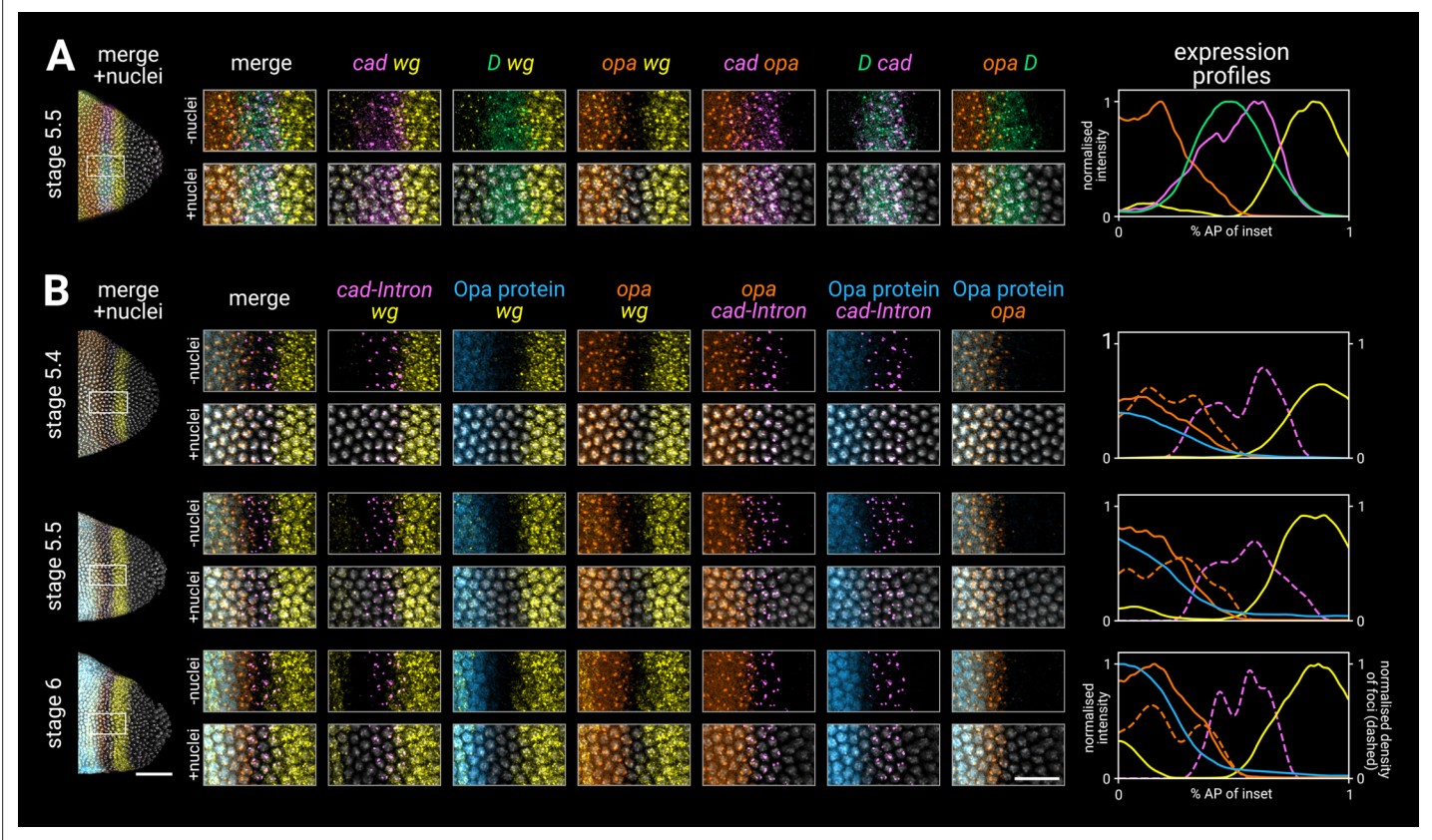

**Figure 3.** Timer gene expression in the tail region of wild-type embryos at high resolution. (**A, B**) Leftmost column shows the posterior ends of the selected embryos, each with a boxed region of interest in the tail; middle columns show high-resolution close-ups of the boxed region without and with DAPI signal ('-nuclei' vs. '+nuclei'); rightmost column shows quantitative expression traces along the x-axis of the boxed region. (**A**) Timer gene expression, as in *Figure 2*. (**B**) *wg* and *opa* expression (as in **A**), combined with a *cad* intronic probe (*cad-Intron*, showing intranuclear transcription foci) and an antibody stain for Opa protein. Solid lines in the expression plots show the average intensity of *wg*, *opa*, and Opa protein; dashed lines show the normalised density of *cad* and *opa* transcription foci. Note the staggered AP distributions of Opa protein, *opa* transcript, and *opa* transcription foci, the shrinking gap between the posterior *wg* domain and the *opa*/Opa signal, and the refinement of the *cad-Intron* domain over time. All embryos are anterior left, dorsal up, lateral view. Scale bars = 50 µm (embryo posteriors), 20 µm (boxed close-ups). For the high-resolution close-ups, the curvature of the tissue was straightened prior to z-projection.

The online version of this article includes the following figure supplement(s) for figure 3:

**Figure supplement 1.** Single-channel images.

At stage 5.4, *cad* is actively transcribed in a domain 3–4 cells wide, but this shrinks to 2–3 cells wide by stage 6, with transcription ceasing at the anterior edge (*cad* intronic probe, *Figure 3B*). Throughout this period, the domain of active *opa* transcription, marked by prominent intranuclear foci, extends about one cell row posterior to the Opa protein domain (*Figure 3B*, Opa/*opa* merge), and also overlaps the *cad* domain by about one cell row (*Figure 3A*, *cad*/*opa* merge; *Figure 3B*, *opa*/*cad-Intron* merge). This suggests that *opa* transcription gradually invades the *cad* tail domain from the anterior edge, with *cad* transcription then ceasing in these cells as Opa levels increase (*Figure 3B*, Opa/*cad-intron* merge). Supporting this interpretation, we confirmed that a posterior expansion of Opa expression is evident in published live-imaging data (*Soluri et al., 2020*).

## Summary

We find that timer gene expression differs sharply between the trunk and the tail, although both regions express *cad*, *D*, and *opa* in the same temporal sequence. The difference in timer gene expression between the trunk and the tail correlates with the difference in simultaneous versus sequential segmentation dynamics described above.

## The timer genes are patterned by cross-regulation

The relative spatiotemporal expression dynamics of the timer genes are suggestive of cross-regulation. To investigate this possibility, we examined timer gene expression in $opa^-$, $D^-$, and $cad^-$ mutants (*Figure 4*; *Figure 4—figure supplement 1*) and discovered a variety of cross-regulatory effects. As *cad* is expressed maternally as well as zygotically, we examined *cad* maternal mutants ($cad^{m-z+}$) and *cad* zygotic mutants ($cad^{m+z-}$) in addition to *cad* null mutants ($cad^{m-z-}$) in order to disentangle maternal and zygotic effects (*Figure 4—figure supplement 3*). We also examined timer gene expression in $wg^-$ mutants, but did not observe any aberrant expression in these embryos during our stages of interest (*Figure 4—figure supplement 4*).

### Timer gene expression in $opa^-$ mutants

In $opa^-$ mutants, trunk expression of *D* persisted longer than usual, resulting in a more prominent stripy pair-rule pattern, while the tail domain was stronger and extended further anterior than normal (*Figure 4A and B*; *Figure 4—figure supplement 1B*; *Figure 4—figure supplement 5*). The *cad* tail domain looked similar to wild-type at stage 5.5 (*Figure 4A and B*; *Figure 4—figure supplement 1B*), but was broader at stage 6 (*Figure 4—figure supplement 5*), suggesting that it failed to retract posteriorly as in wild-type. *opa* transcription and the posterior *wg* domain looked normal.

### Timer gene expression in $D^-$ mutants

In $D^-$ mutants, *cad* expression persisted abnormally in the trunk, with marked AP modulation, and the *cad* tail domain extended further anterior than normal (*Figure 4A–C*; *Figure 4—figure supplement 1B*). The *D* allele we used had very low transcript levels (presumably due to nonsense-mediated decay, S. Russell pers. comm.), but the residual expression indicated that both the clearance of *D* expression from the trunk and the appearance of the *D* tail domain may have been delayed. The posterior *wg* domain, the posterior border of the *cad* tail domain, and the posterior border of the *opa* domain were all modestly anteriorly shifted relative to wild-type (*Figure 4B*; *Figure 4—figure supplement 1A*; *Figure 4—figure supplement 2*); even after allowing for this shift, the gap between the *wg* domain and the *opa* domain was slightly larger in $D^-$ embryos than in wild-type (*Figure 4—figure supplement 1B*).

### Timer gene expression in $cad^{m-z-}$ mutants

In $cad^{m-z-}$ mutants, *cad* expression persisted abnormally in the trunk (*Figure 4A–C*), though without the AP modulation seen in $D^-$ mutants. *D* expression levels were weaker than normal at early stage 5 (*Figure 4C*, stage 5.2), the *D* neuroectodermal expression domain appeared precociously (*Figure 4C*, stage 5.4), and the *D* tail domain was only expressed in the ventral half of the embryo (arrowhead in *Figure 4A*). The posterior *wg* domain was generally absent (arrowhead in *Figure 4A*; *Wu and Lengyel, 1998*), although weak expression was observed in some embryos, consistent with the variability of the $cad^{m-z-}$ larval phenotype (*Macdonald and Struhl, 1986*). The *opa* domain showed strong pair-rule modulation in the anterior trunk (arrowheads in *Figure 4A*; *Figure 4—figure supplement 2*).

### Timer gene expression in $cad^{m+z-}$ and $cad^{m-z+}$ mutants

One copy of maternal *cad* ($cad^{m+z-}$ embryos) largely rescued the $cad^{m-z-}$ phenotype, except that the *D* tail domain was lost prematurely, during germband extension (*Figure 4—figure supplement 3B*). The posterior *wg* domain was present, conflicting with a previous report (*Wu and Lengyel, 1998*).

One copy of zygotic *cad* ($cad^{m-z+}$ embryos) rescued the *D* tail domain fully and partially rescued the *wg* posterior domain (*Figure 4—figure supplement 3C*), but the blastoderm dynamics of *D* and *cad* expression were still perturbed.

### Other observations from $cad^{m-z-}$ mutants

We wondered whether the premature neuroectodermal expression of *D* in $cad^{m-z-}$ mutants might indicate a more general pattern of precocious neuroectoderm development. To investigate this, we examined the expression of *muscle segment homeobox* (*msh*, also known as *Drop*; *Lord et al., 1995*), a key neuroectoderm patterning gene expressed outside the *D* neuroectodermal domain. We found

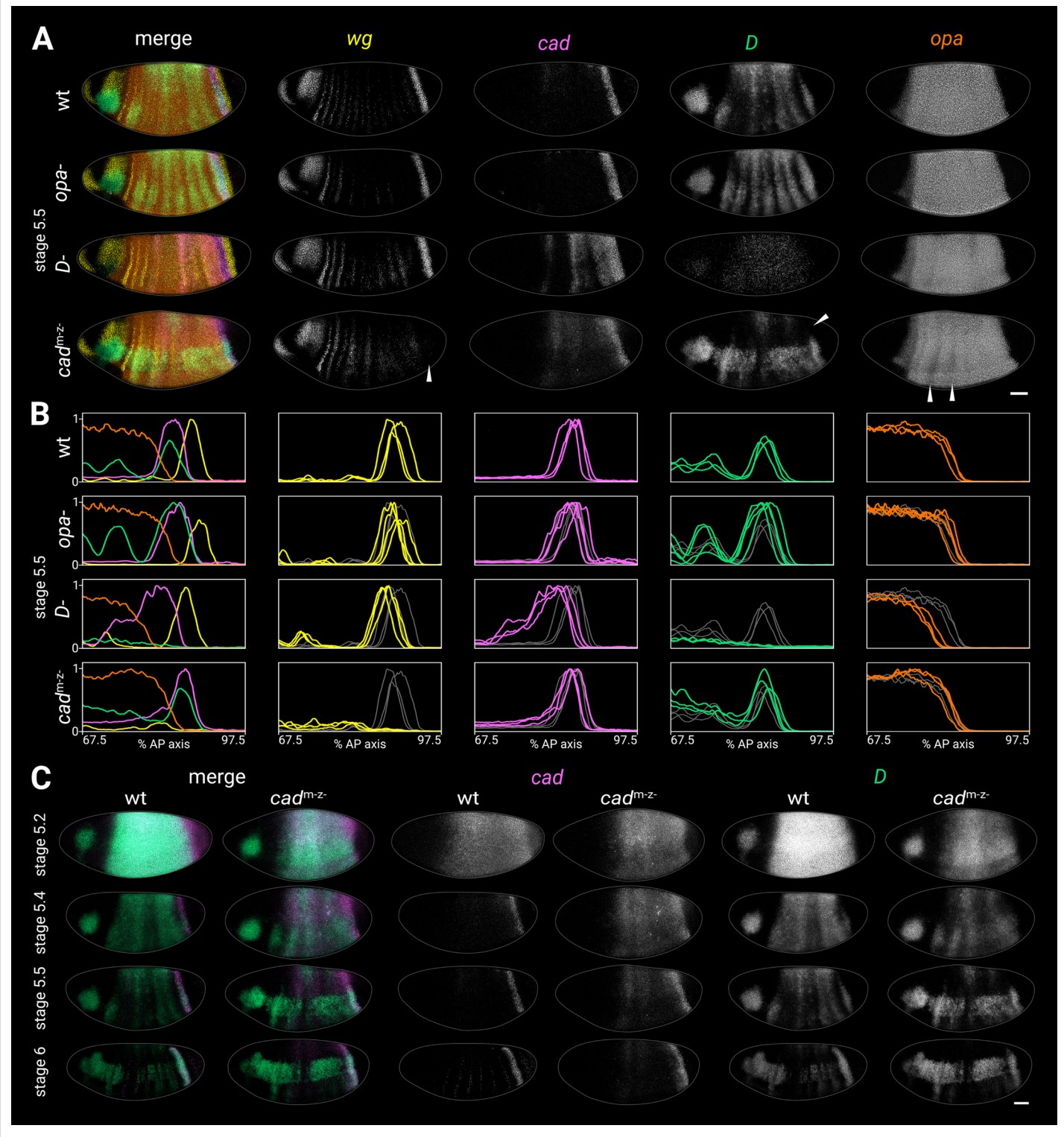

**Figure 4.** Timer gene expression in timer gene mutants. (**A**) Timer gene expression in wild-type, *opa⁻* mutants, *D⁻* mutants, and *cad*^m-z-^ mutants at stage 5.5. The leftmost column shows a four-channel merge and the other columns show individual channels. In the *cad*^m-z-^ embryo, note the absence of the *wg* posterior domain (arrowhead in *wg* channel), the dorsal loss of the *D* tail domain (arrowhead in *D* channel), and the AP modulation of the *opa* trunk domain (arrowheads in *opa* channel). The brightness and contrast of the *D* channel were adjusted for the *D⁻* embryo to reveal the very weak residual signal. (**B**) Quantitative expression traces (67.5–97.5% AP axis) from the individual embryos in (**A**) (multi-channel traces in leftmost column) or multiple stage 5.5 embryos (single-channel traces in other columns). All traces are individually normalised; mutant traces are overlaid on wild-type traces (grey)

*Figure 4 continued on next page*

*Figure 4 continued*

for ease of comparison. (**C**) *cad* and *D* expression in wild-type and *cad*<sup>m-z-</sup> mutant embryos of gradually increasing age; leftmost columns show a two-channel merge. In the *cad*<sup>m-z-</sup> embryos, note that *cad* transcript takes longer to clear from the trunk, while *D* is initially expressed at lower intensity and its neuroectodermal expression domain emerges earlier. All embryos are anterior left, dorsal up, lateral view. Scale bar = 50 µm; grey lines show embryo outlines.

The online version of this article includes the following source data and figure supplement(s) for figure 4:

**Source data 1.** Expression trace source data.

**Figure supplement 1.** Timer gene expression traces from timer gene mutants, relative to wild-type.

**Figure supplement 2.** Full-length *opa* expression traces from timer gene mutants.

**Figure supplement 3.** Additional characterisation of *cad*<sup>m-z-</sup>, *cad*<sup>m-z+</sup>, and *cad*<sup>m+z-</sup> mutants.

**Figure supplement 4.** Timer gene expression in *wg*<sup>-</sup> mutants at stage 6.

**Figure supplement 5.** Timer gene expression in *opa*<sup>-</sup>, *tll*<sup>-</sup>, and *tll*<sup>-</sup> *opa*<sup>-</sup> embryos at stage 6.

**Figure supplement 6.** Morphological differences between wild-type and *cad*<sup>m-z-</sup> blastoderms.

**Figure supplement 6—source data 1.** Measurements of wild-type and *cad*<sup>m-z-</sup> blastoderms.

that *msh* was also expressed prematurely in *cad*<sup>m-z-</sup> mutants, particularly in posterior parts of the embryo (***Figure 4—figure supplement 3A***).

Fixed and mounted *cad*<sup>m-z-</sup> embryos had a different range of shapes and sizes compared to wild-type embryos (***Figure 4—figure supplement 6***). We did not investigate whether this was specifically due to the loss of Cad expression or an artefact of the 'FLP-DFS' technique for generating germline clones (***Chou and Perrimon, 1996***). Given the robustness of AP patterning to variation in embryonic geometry (***Huang et al., 2020***), this minor morphological effect is unlikely to be the cause of the gene expression changes we observed.

## Summary

Our investigation of timer gene mutant phenotypes provides strong evidence for timer gene cross-regulation. *cad* is derepressed in *D*<sup>-</sup> mutants, and *D* is derepressed in *opa*<sup>-</sup> mutants. *cad*<sup>m-z-</sup> embryos have a complex phenotype in which the early expression of *D* is reduced, neuroectodermal gene expression is activated prematurely, the posterior *wg* domain is lost, and the *D* tail domain fails to activate dorsally. Finally, *opa* expression is fairly normal across all the mutants, except that its posterior border is anteriorly shifted in *D*<sup>-</sup> mutants.

These phenotypes, in combination with the expression dynamics described in the previous section, suggest that Opa represses *D* and *cad*, D represses *cad*, and Cad activates *D* (see ***Appendix 3—table 1*** for detailed reasoning). In addition, Cad is required for the expression of posterior *wg*, and D has a modest but concerted effect on the entire posterior fate map. Finally, most of the *cad*<sup>m-z-</sup> phenotype is mediated by maternal Cad, but zygotic Cad has specific late effects on *D* in the tail.

## Tll and Hkb expression dynamics correlate with timer gene patterning in the posterior of the embryo

We next wanted to understand why timer gene expression differs between the trunk, tail, and prospective gut regions; i.e., how the timer gene network is spatially regulated. We therefore examined how timer gene expression relates to the expression domains of the zygotic terminal system genes *tll* (***Jurgens et al., 1984***; ***Strecker et al., 1986***; ***Pignoni et al., 1990***) and *huckebein* (*hkb*; ***Weigel et al., 1990***; ***Brönner and Jäckle, 1991***), the obvious candidates for providing this spatial information.

### *tll* and *hkb* expression dynamics

*tll* and *hkb*, which both code for repressive transcription factors, are expressed in nested domains at the posterior pole, with *tll* expression extending further from the pole than *hkb* expression (***Figure 5***; ***Figure 5—figure supplement 1***; ***Pignoni et al., 1990***; ***Brönner and Jäckle, 1991***). *tll* is transcribed at low levels from as early as nuclear cycle 9 (***Pignoni et al., 1992***), and we detected similar early transcription for *hkb*. Transcript levels in both domains peak at around stage 5.2 and then decline, with *tll* expression fading by stage 6 and *hkb* persisting at low levels after gastrulation (***Figure 5***; ***Figure 5—figure supplement 1***; ***Figure 5—figure supplement 3***). Previous studies (***Pignoni et al.,***

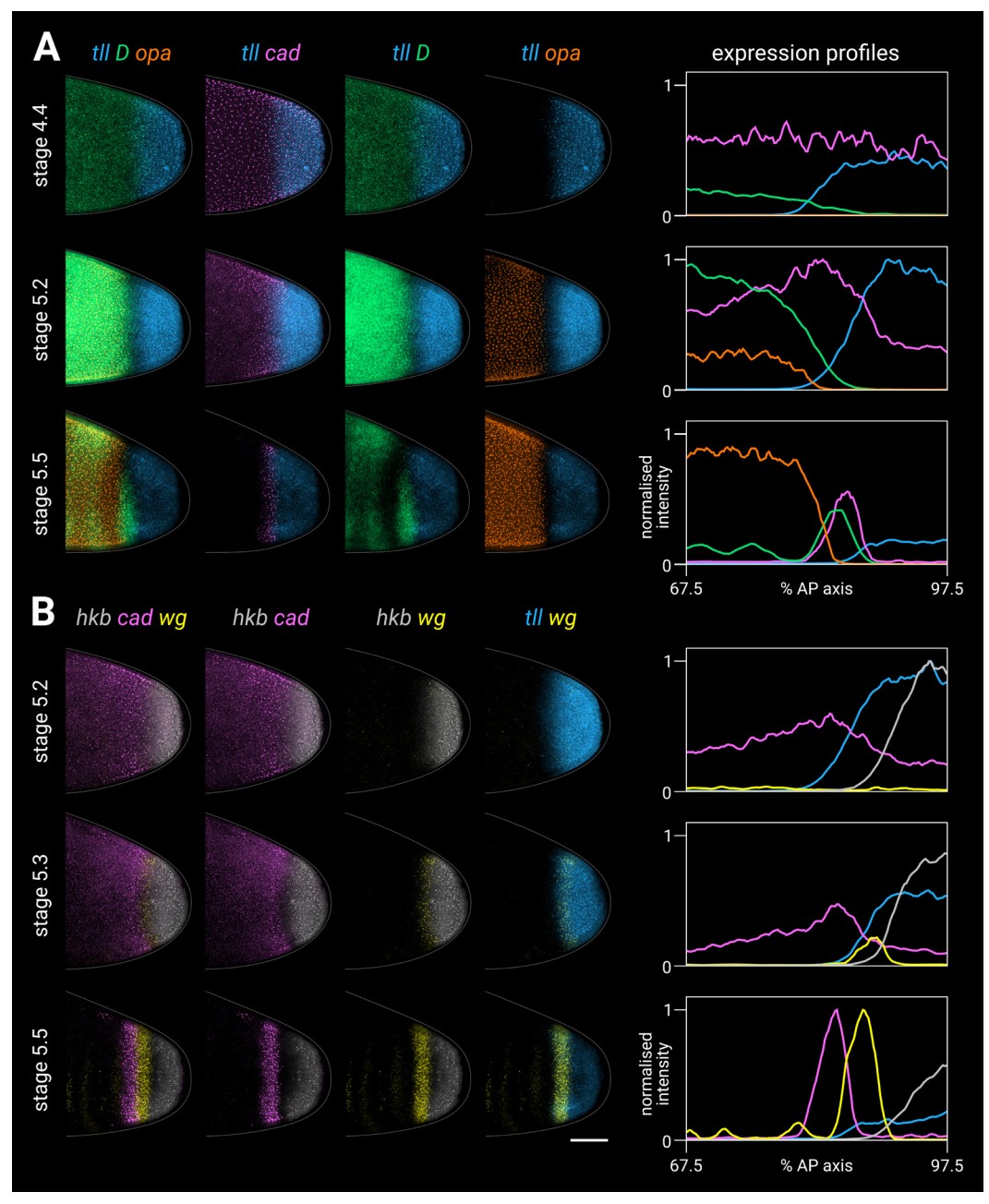

**Figure 5.** Timer gene expression relative to posterior terminal gene expression in wild-type embryos. (**A, B**) Timer and terminal gene expression in embryos of increasing age; only the posterior end of each embryo is shown. Left four columns show either three-channel or two-channel merges; right column shows quantitative expression traces (67.5–97.5% AP axis) of all four genes in the stain. (**A**) Timer gene expression relative to *tll*; note the posterior regression and changing intensity of the *tll* domain and the different spatial relationships with *opa*, *D*, and *cad*. (**B**) *cad* and *wg* expression relative to *hkb* and *tll*; note how the posterior *wg* domain emerges within the *tll*-positive gap that opens up between *cad* and *hkb*. All embryos are anterior left, dorsal up, lateral view. Scale bar = 50 μm; grey lines show embryo outlines.

The online version of this article includes the following source data and figure supplement(s) for figure 5:

**Source data 1.** Expression trace source data.

**Source data 2.** Measurements of the size of the posterior *tll* expression domain at stage 5.2 vs. 5.5.

**Figure supplement 1.** Expression of *tll* and *hkb* in wild-type embryos from stage 2 to stage 6.

**Figure supplement 2.** Single-channel images.

*Figure 5 continued on next page*

**Figure supplement 4.** Relative expression dynamics of Tll protein and *tll* transcript.

**Figure supplement 5.** Relative expression dynamics of Hkb, Tll, and Opa proteins.

**Figure supplement 6.** Examples of source imaging data for supplementary plots.

*1990*; *Pignoni et al., 1992*) reported retraction of the *tll* border by about 5% egg length between stage 4.4 (nuclear cycle 13) and stage 5 (nuclear cycle 14); we noticed that this border also retracts by about 3–4 nuclear diameters over the course of stage 5 (*Figure 5—source data 2*). (Note that the absolute [% AP axis] shifts in *Figure 5—figure supplement 3* appear smaller than this because the posterior retraction of gene expression across nuclei is partially cancelled out by the anterior flow of nuclei away from the pole; *Keränen et al., 2006*.)

Tll and Hkb protein dynamics (*Figure 5—figure supplement 4*; *Figure 5—figure supplement 5*) are spatiotemporally similar to *tll/hkb* transcript dynamics, albeit with a slight time lag, with the Tll protein border therefore lying slightly anterior to the *tll* transcript border during the second half of stage 5 (*Figure 5—figure supplement 4A*). Our Tll antibody data closely resembles that collected by the Reinitz group, who noted that "*in contrast to the posterior domains of the other gap genes, the [Tll] posterior domain does not shift position with time*" (*Surkova et al., 2008*). We interpret the same data as providing evidence for a modest posterior retraction of the Tll domain over time, which does indeed contrast with the anterior shifts of the trunk gap genes, and is partially masked by anterior nuclear flow.

### *tll* and *hkb* expression dynamics relative to the timer genes

The *tll* and *hkb* anterior borders correlate closely with the resolving expression boundaries of *cad*, *D*, *opa*, and *wg* (*Figure 5*). At stage 4.4 (nuclear cycle 13), the graded *tll* border overlaps the graded posterior edge of the *D* domain (*Figure 5A*, top row). By mid stage 5, a narrow gap of low expression opens between the *tll* domain and the trunk domains of *D* and *opa* (*Figure 5A*, middle row), which is then filled by the *cad* and *D* tail domains at late stage 5 (*Figure 5A*, bottom row). *cad* is expressed ubiquitously throughout the posterior of the embryo at stage 4.4 (*Figure 5B*, top row), then fades from the *hkb* domain by mid stage 5 (*Figure 5B*, middle row), with a narrow gap of low expression opening up between the *cad* and *hkb* domains by late stage 5 (*Figure 5B*, bottom row). The *wg* posterior domain initiates at the border between *cad* and *hkb* expression present at mid stage 5 (*Figure 5B*, middle row), and by late stage 5 the *wg* posterior domain neatly demarcates the strip of *tll*-positive *hkb*-negative cells (*Figure 5B*, bottom row).

### Summary

The spatiotemporal expression dynamics of Tll and Hkb make them good candidates for patterning the timer gene boundaries and the posterior *wg* domain because they are differentially expressed across the various terminal regions. Specifically, from posterior to anterior, the prospective posterior midgut experiences strong expression of both Tll and Hkb, the prospective hindgut experiences strong expression of Tll but weak/transient expression of Hkb, the tail region experiences weak/transient expression of Tll, and the trunk is consistently free of Tll and Hkb expression.

## The terminal system interacts with the timer gene network to pattern the posterior of the embryo

To determine whether Hkb and Tll spatially regulate the timer genes, we investigated timer gene expression in *hkb*⁻ mutants, *tll*⁻ mutants, and *torso* (*tor*) mutants (*Figure 6*). Tor (*Klingler et al., 1988*; *Sprenger et al., 1989*; *Casanova and Struhl, 1989*) is a maternally provided receptor necessary for transducing the extracellular signal-regulated kinase (ERK) signal that specifies the poles of the embryo (reviewed in *Duffy and Perrimon, 1994*; *Li, 2005*; *Goyal et al., 2018*), and therefore *tor*⁻ mutants express neither *hkb* nor *tll* (*Brönner and Jäckle, 1991*; *Pignoni et al., 1992*).

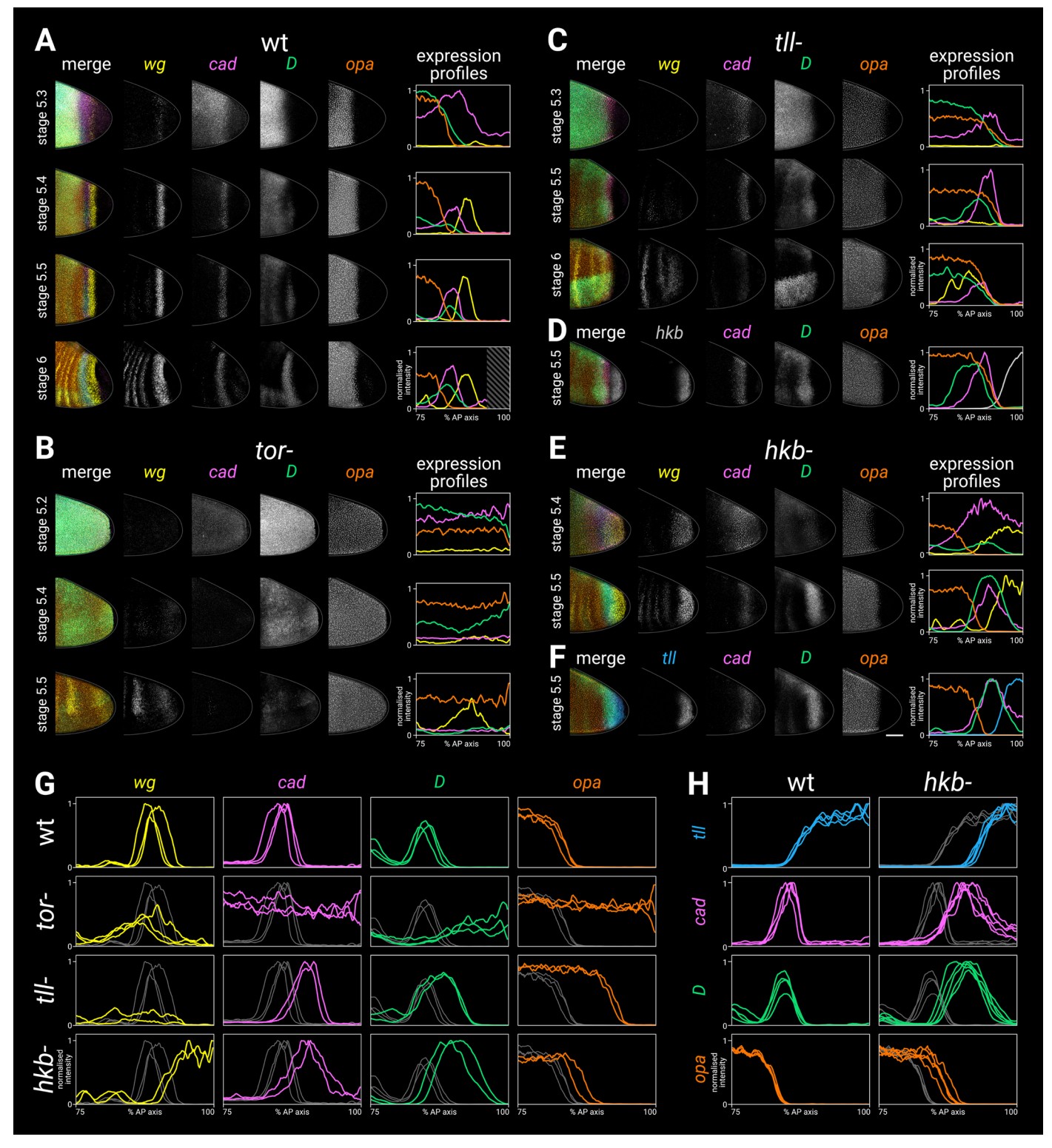

**Figure 6.** Timer gene expression in terminal system mutants. (**A–F**) Gene expression in wild-type and mutant embryos of increasing ages. The leftmost column shows a four-channel merge; the middle columns show individual channels; the rightmost column shows quantitative expression traces (75–100% AP axis) from the embryos shown to the left. (**A**) Timer gene expression in wild-type. The AP axis is truncated in the expression plot for the stage 6 embryo (diagonally shaded area) due to proctodaeal invagination. (**B**) Timer gene expression in *tor⁻* mutants. Note how the timer gene expression expands all the way to the posterior pole (excluding the pole cells). The broad posterior *wg* domain seen at stage 5.4–5.5 is mispatterned

*Figure 6 continued on next page*

Figure 6 continued

segmental expression; the posterior *wg* domain seen in wild-type embryos is absent. (**C**) Timer gene expression in *tll*‑ mutants, relative to *wg* expression. Note that the *cad*, *D*, and *opa* domains share a similar posterior border, the *cad* domain fades over time, and the *wg* posterior domain is absent. (Some mispatterned segmental *wg* expression is seen near the posterior of the embryo, similar to *tor*‑ mutants.) (**D**) Timer gene expression in *tll*‑ mutants, relative to *hkb* expression. Note that the posterior borders of *cad*, *D*, and *opa* all abut the *hkb* expression domain. (**E**) Timer gene expression in *hkb*‑ mutants, relative to *wg* expression. Note that *cad* is not repressed from the posterior pole until stage 5.5, and the posterior *wg* domain extends to the posterior pole. (**F**) Timer gene expression in *hkb*‑ mutants, relative to *tll* expression. Note that the *tll* domain is small, and it preserves normal relationships with the *cad*, *D*, and *opa* domains. (**G, H**) Single-channel quantitative expression traces (75–100% AP axis) from multiple wild-type and mutant stage 5.5 embryos. Note the absence of spatial patterning in *tor*‑ mutants and the posteriorly shifted expression boundaries in *tll*‑ and *hkb*‑ mutants. In (**A–F**) all embryos are anterior left, dorsal up, lateral view; scale bar = 50 µm; grey lines show embryo outlines. In (**G,H**) all traces are individually normalised; mutant traces are overlaid on wild-type traces (grey) for ease of comparison.

The online version of this article includes the following source data and figure supplement(s) for figure 6:

**Source data 1.** Expression trace source data.

**Figure supplement 1.** Timer gene expression traces from terminal system mutants, relative to wild-type.

**Figure supplement 2.** Expression of *tll* in wild-type and *hkb*‑ embryos from stage 5.5 to stage 7.

## Timer gene expression in *tor*‑ mutants

In *tor*‑ mutants (**Figure 6B and G**), all posterior spatial patterning of the timer genes was lost, and their temporal expression dynamics resembled those seen in the trunk of wild-type embryos. Thus *cad*, *D*, and *opa* were all expressed to the very posterior of the embryo at the beginning of stage 5, with first *cad* and then *D* expression turning off as stage 5 progressed. The posterior domain of *wg* was absent, and the region of segmental *wg* expression expanded posteriorly, as described previously (**Mohler, 1995**). Loss of the *cad* tail domain in *tor*‑ and *torso-like* (*tsl*‑) mutants has also been described previously (**Mlodzik and Gehring, 1987b**; **Schulz and Tautz, 1995**).

## Timer gene expression in *tll*‑ and *tll*‑ *opa*‑ mutants

In *tll*‑ mutants (**Figure 6C, D and G**; **Figure 6—figure supplement 1A**), the posterior *wg* domain was absent (**Wu and Lengyel, 1998**), and the *cad*, *D*, and *opa* domains were expanded posteriorly to abut the *hkb* domain, which looked similar to wild-type (**Figure 7B**). Normal expression of *hkb* in *tll*‑ mutants has been previously reported (**Brönner and Jäckle, 1991**; **Brönner et al., 1994**; **Ashyraliyev et al., 2009**).

A posteriorly shifted *cad* tail domain was transiently expressed (**Figure 6C and G**; **Figure 4—figure supplement 5**). This finding conflicts with previous reports that the *cad* tail domain was either unaffected (**Reinitz and Levine, 1990**) or completely absent (**Mlodzik and Gehring, 1987b**) in *tll*‑ mutants.

The pattern of *D* expression in the trunk was abnormal (presumably caused by feedback from the segmentation genes, which are misregulated in *tll*‑ mutants; **Mahoney and Lengyel, 1987**; **Casanova, 1990**; **Janssens et al., 2013**), and a persistent posterior *D* domain did not emerge (**Figure 6C**; **Figure 4—figure supplement 5**).

*tll*‑ *opa*‑ double mutants showed similar patterning dynamics to *tll*‑ single mutants, except that tail-like expression of *D* was rescued and persisted into germband extension (**Figure 4—figure supplement 5**).

## Timer gene expression in *hkb*‑ mutants

In *hkb*‑ mutants (**Figure 6E–H**), the *wg* posterior stripe became a posterior cap (**Mohler, 1995**), and *cad* expression persisted longer than normal at the posterior pole. The relative phasing of the *cad*, *D*, *opa*, and *wg* domains was preserved, but the whole terminal pattern was posteriorly shifted/expanded into territory that would normally express *hkb* (**Figure 6H**).

In contrast to previous reports that *tll* expression is unaffected in *hkb*‑ mutants (**Brönner and Jäckle, 1991**; **Brönner et al., 1994**; **Brönner and Jäckle, 1996**), we found that the *tll* domain was smaller than normal, thereby preserving the correlation between *tll* levels and timer gene expression boundaries seen in wild-type embryos (**Figure 6F and H**; **Figure 6—figure supplement 1B**). Expression of *tll* persisted throughout stages 6 and 7, rather than fading at stage 6, and ectopic expression appeared at the anterior pole (**Figure 6—figure supplement 2**).

## Summary

All posterior spatial patterning of the timer genes is dependent on the terminal system via *tor*. Expression boundaries associated with the tail and hindgut are perturbed in *tll⁻* mutants, while expression boundaries associated with the posterior midgut are perturbed in *hkb⁻* mutants. In addition, there is a concerted posterior shift of the fate map in *hkb⁻* mutants, which we attribute to the reduced size of the *tll* domain.

Our observations from this and the previous section suggest that Tll strongly represses *D* and *opa* and weakly represses *cad*, while Hkb represses *wg*, *cad*, *D*, and *opa* (see **Appendix 3—table 1** for detailed reasoning). Hkb is also necessary for activation of *tll* at normal levels (an interaction that is presumably indirect since Hkb acts as a repressor; *Goldstein et al., 1999*), and for timely repression of *tll* after stage 5.

## Fkh demarcates the tail/hindgut border and activates posterior *wg*

Having found that Tll is necessary for patterning both the tail region and the posterior *wg* domain (prospective hindgut), we next asked how these regions are distinguished from each other. Forkhead (Fkh) is a zygotic transcription factor that is expressed in the posterior of the embryo from stage 4.4 (nuclear cycle 13) downstream of Tor (*Weigel et al., 1989*; *Weigel et al., 1990*) and is required for the specification of hindgut identity (*Jürgens and Weigel, 1988*; *Weigel et al., 1989*; *Kuhn et al., 1995*; *Hoch and Pankratz, 1996*).

### *fkh* expression in *cad*^m-z-^, *hkb⁻*, and *tll⁻* mutants

We examined the expression of *fkh* relative to other terminal genes in wild-type embryos and in mutant genotypes in which tail or hindgut patterning is perturbed (**Figure 7A and B**).

In wild-type embryos at stage 5.4, the posterior *fkh* domain had a fairly sharp border, which lined up with the anterior border of the posterior *wg* domain and the posterior border of the *cad* tail domain.

In *cad*^m-z-^ mutants, *fkh* expression was strongly reduced (*Wu and Lengyel, 1998*), contrasting with the *tll* and *hkb* domains in these embryos, which looked normal (**Figure 7—figure supplement 1**; *Wu and Lengyel, 1998*; *Olesnicky et al., 2006*).

In *hkb⁻* mutants, the *fkh* domain was reduced in size (*Weigel et al., 1990*; *Gaul and Weigel, 1990*), correlating with the reduced size of the *tll* domain and the posteriorly shifted *wg* and *cad* borders in this genotype.

The *fkh* domain was also reduced in *tll⁻* mutants (*Weigel et al., 1990*; *Gaul and Weigel, 1990*). The reduced domain was the same size as the *hkb* domain, and it abutted the posteriorly shifted *cad* tail domain.

### Timer gene expression in *fkh⁻* mutants

In *fkh⁻* mutants (**Figure 7C and D**), the posterior *wg* domain was largely absent (*Wu and Lengyel, 1998*), although there was some residual posterior *wg* expression, particularly in ventral tissue. *cad*, *D*, and *opa* expression was essentially normal throughout stage 5, although the *cad* posterior border appeared to be slightly posteriorly expanded relative to the *D* tail domain.

A stronger effect on *cad* expression was seen after gastrulation, when new *cad* transcription appeared posteriorly abutting the *cad* tail domain, rather than several cells away (posterior to *wg*) as in wild-type embryos (**Figure 7—figure supplement 3**). Our findings contrast with a previous report, which described *cad* expression as being normal in *fkh⁻* mutants (*Jürgens and Weigel, 1988*).

### Abnormal morphogenesis in *fkh⁻* and *cad*^m-z-^ mutants

Morphogenesis was abnormal in *fkh⁻* mutants, in that proctodaeal invagination was delayed until after stage 7 (**Figure 7—figure supplement 4**). This finding contrasts with previous reports that morphogenesis in *fkh⁻* mutants is normal until the end of the extended germband stage (*Weigel et al., 1989*; *Wu and Lengyel, 1998*).

*cad*^m-z-^ mutants (which have severely reduced *fkh* expression) show a similar morphogenetic delay (**Figure 7—figure supplement 4**) as well as other defects in posterior invagination (*Wu and Lengyel, 1998*). Posterior invagination is dependent on Fog signalling (*Costa et al., 1994*; *Sweeton*

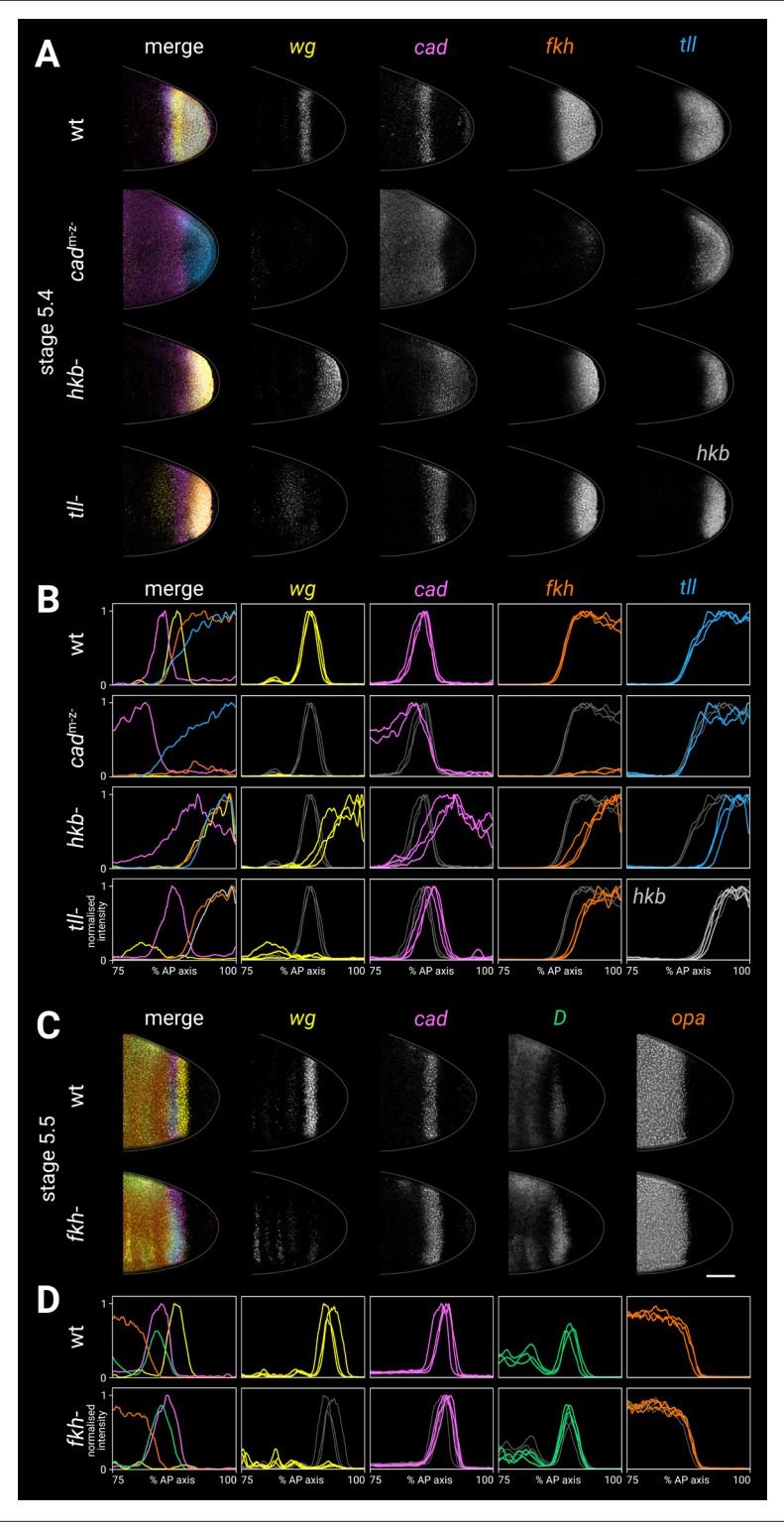

**Figure 7.** Spatial regulation of *fkh*, and timer gene expression in *fkh*⁻ mutants. (**A, B**) Terminal gene expression (*wg*, *cad*, *fkh*, and *tll/hkb*) in wild-type and mutant embryos. In *cad*^m-z-, note the loss of *wg* and *fkh* expression. In *hkb*⁻, note the posterior fate map shift and the delayed repression of posterior *cad*. In *tll*⁻, note the loss of the posterior *wg* domain, the posteriorly shifted *cad* domain, and the reduced size of the *fkh* domain. (**C, D**) Timer gene expression in wild-type and *fkh*⁻ mutant embryos. Note the extremely reduced posterior *wg* domain in *fkh*⁻. (**A, C**) Individual stage 5.4 (**A**) or stage 5.5 (**C**) embryos; the leftmost column shows a four-channel merge, other columns

*Figure 7 continued on next page*

*Figure 7 continued*

show individual channels. All embryos are anterior left, dorsal up, lateral view. Scale bar = 50 μm; grey lines show embryo outlines. (**B, D**) Quantitative expression traces (75–100% AP axis); the leftmost column shows multi-channel traces from the individual embryos in (**A, C**), other columns show single-channel traces from multiple stage 5.4–5 embryos (**B**) or stage 5.5 embryos (**D**). All traces are individually normalised; mutant traces are overlaid on wild-type traces (grey) for ease of comparison.

The online version of this article includes the following source data and figure supplement(s) for figure 7:

**Source data 1.** Expression trace source data.

**Figure supplement 1.** Terminal gene expression traces from *cad*^m-z-^, *tll*^-^, and *hkb*^-^ mutants, relative to wild-type.

**Figure supplement 2.** Timer gene expression traces from *fkh*^-^ mutants, relative to wild-type.

**Figure supplement 3.** *wg* and *cad* expression in *fkh*^-^ mutants at stages 6 and 7.

**Figure supplement 4.** Abnormal morphogenesis in *cad*^m-z-^ and *fkh*^-^ mutants.

---

*et al., 1991*; *Parks and Wieschaus, 1991*), which is known to be reduced in *cad*^m-z-^ mutants (*Wu and Lengyel, 1998*). As Fkh is known to activate Fog signalling in other developmental contexts (*Chung et al., 2017*), the reduction in Fog signalling may be mediated by the reduction in Fkh.

## Summary

We found a consistent pattern across wild-type, *cad*^m-z-^, *hkb*^-^, and *tll*^-^ genotypes, in which the *fkh* border abutted the posterior border of the *cad* tail domain, and posterior *wg* was only expressed in *fkh*-positive *hkb*-negative territory. Accordingly, in *fkh*^-^ mutants, the posterior *wg* domain was largely lost.

These results are consistent with previously proposed regulatory interactions: that Fkh activates *wg* (*Wu and Lengyel, 1998*), that Cad activates *fkh* (*Wu and Lengyel, 1998*), and that Tll and Hkb indirectly enable *fkh* to be expressed (*Weigel et al., 1990*; *Casanova, 1990*; *Goldstein et al., 1999*; *Morán and Jiménez, 2006*). Accordingly, the activation of *wg* by Cad (*Wu and Lengyel, 1998*) appears to be indirect, via Fkh (see *Appendix 3—table 1* for detailed reasoning). In addition, it is possible that Fkh represses *cad*, but current evidence is inconclusive (see *Appendix 3—table 1*).

## Inferred regulatory interactions collectively form a network that can be formalised and simulated

From looking at how gene expression is affected in various mutant genotypes, we have inferred a network of regulatory interactions between the timer genes and the posterior terminal genes (*Figure 8A*; *Appendix 3—table 1*). Most (11/18) of these proposed interactions originate from this study, although we also find support for previously proposed interactions related to the patterning of *tll*, *hkb*, *fkh,* and *wg* (*Figure 8B*). (For a recent quantitative model of posterior gut specification using a network similar to *Figure 8B*, see *Keenan et al., 2022*.)

We now formalise the regulatory network in *Figure 8A* as a logical model, and see whether it reproduces the patterning dynamics that we observed in the embryo. For the purposes of this study, we are aiming for a minimal, qualitative explanation of timer gene patterning, commensurable with the essentially qualitative developmental genetic paradigm we have been working within. We are interested in the relative ordering of gene expression domains in time and space, abstracted away from specific domain sizes, expression levels or expression kinetics. To the extent that the model is able to recapitulate the essential features of both wild-type and mutant genotypes, our confidence in the network will be increased.

The modelling framework we have chosen is very simple (for a full description, see Appendix 4). Briefly, Hkb and Tll are assumed to be extrinsic inputs to the system (we ignore the cross-regulation of *tll* by Hkb), and we model how Fkh, Wg, Cad, D, and Opa are expressed in response. Each of these seven factors is modelled as a logical variable, some of which (`Hkb`, `Tll`, `D`, `Opa`) may take one of three levels of expression (off/weak/strong), while the others (`Fkh`, `Wg`, `Cad`) may take only two (off/on). The AP axis is modelled as four discrete regions, 1–4 (corresponding to trunk, tail, hindgut, and posterior midgut, respectively), which differ in their hard-coded `Hkb` and `Tll` inputs over time. (Note that we do not include any dorsoventral input to the system, nor attempt to model the *D* neuroectodermal

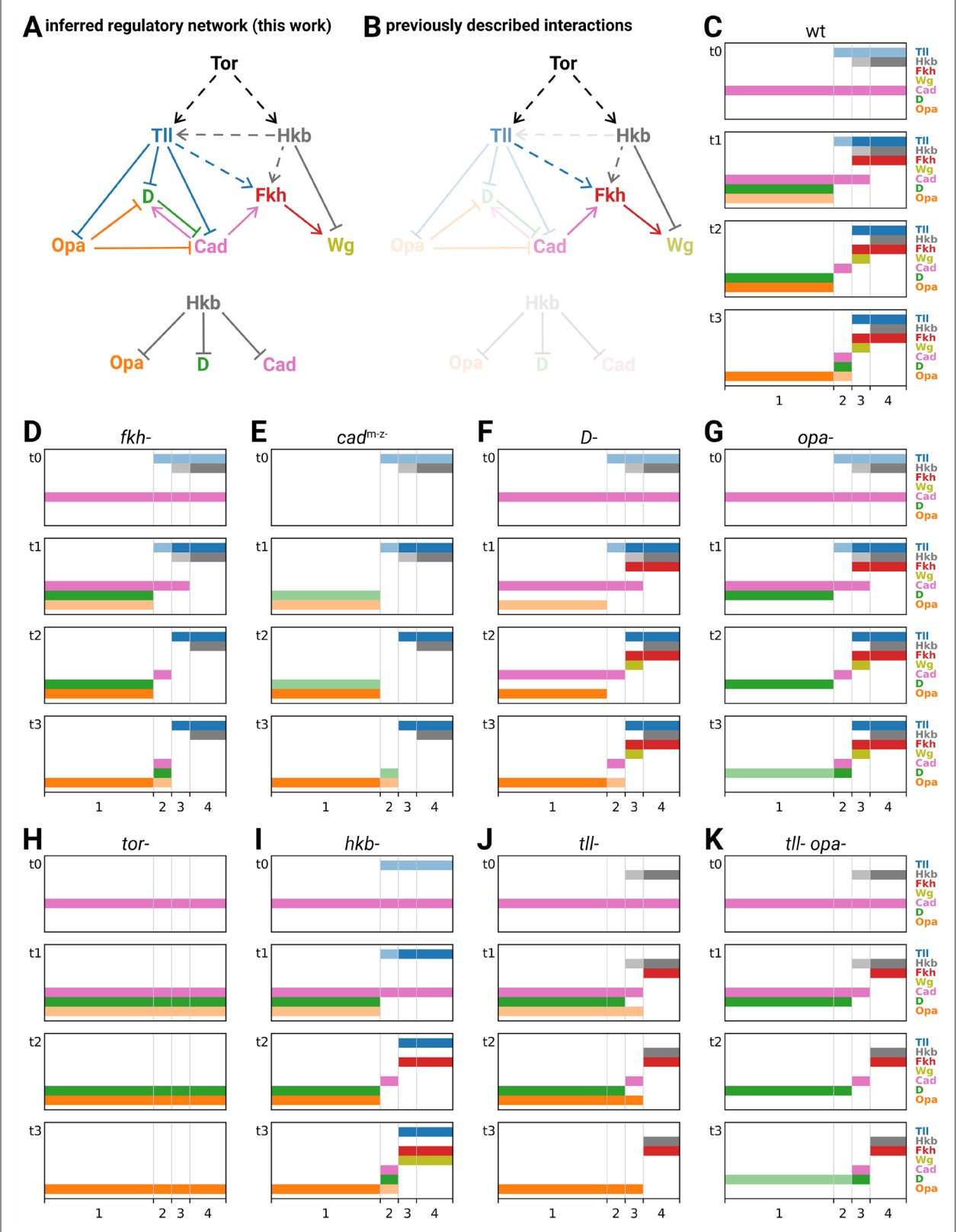

**Figure 8.** Inferred regulatory network for posterior terminal patterning and output of resulting model. (**A**) Arrow diagram showing the regulatory interactions we have inferred from the experiments described in this work. Pointed arrowheads indicate activation; flat arrowheads indicate repression. Solid lines indicate interactions that are presumed to be direct; dashed lines indicate interactions that are presumed to be indirect. The diagram is laid out so that the factors are arranged in approximately the same order left to right as their expression along the AP axis, and causation mainly flows from

*Figure 8 continued on next page*

*Figure 8 continued*

top to bottom (with exceptions for Opa and Cad). To avoid arrow crossovers, the repression of Opa, D, and Cad by Hkb is shown separately from the main network. (**B**) The same network as in (**A**), highlighting the interactions described in the existing literature. (**C–K**) Simulation output for a logical model of posterior terminal patterning, for wild-type and eight mutant genotypes (see main text for details). Each set of plots shows the expression patterns of the logical variables Tll, Hkb, Fkh, Wg, Cad, D, and Opa (y-axis) across AP regions 1–4 (x-axis), at timepoints *t0–t3*. For Tll, Hkb, D, and Opa, a light colour shade represents weak expression and a dark colour shade represents strong expression. Mutant genotypes never express the relevant protein; *tor* mutants were simulated as *tll⁻ hkb⁻* double mutants.

The online version of this article includes the following figure supplement(s) for figure 8:

**Figure supplement 1.** Simulation of a hypothetical timer gene network for sequential segmentation.

domain.) Each simulation consists of four time points, *t0–t3* (corresponding to nuclear cycle 13, early stage 5, mid stage 5, and stage 6, respectively). At *t0*, Cad is on in all regions, and the other output factors are off. Expression at subsequent timepoints is computed from expression at *t(n* − 1), according to factor-specific logical rules (which remain the same for all timepoints). Mutants are simulated by keeping the relevant factor(s) turned off for all timepoints.

## The regulatory network explains the patterning dynamics of each genotype

We simulated the patterning model for the wild-type condition (*Figure 8C*) and eight mutant genotypes examined in this study (*fkh⁻*, *cad^(m-z-)*, *D⁻*, *opa⁻*, *tor⁻*, *hkb⁻*, *tll⁻*, and *tll⁻ opa⁻*; *Figure 8D–K*). A genotype-by-genotype explanation of the simulated expression dynamics is provided in Appendix 4, along with a table cross-referencing the simulated expression data with the corresponding observations from real embryos (*Appendix 4—table 1*). Allowing for the simple, qualitative nature of the model, the simulations were remarkably accurate at recapitulating the patterning dynamics of each genotype.

### Recapitulation of wild-type patterning

Regions 1–4 generate different gene expression as a result of their different inputs from Tll and Hkb. Across regions 3 and 4, the nested domains of strong Tll and Hkb expression specify abutting domains of hindgut (Fkh and Wg) and posterior midgut (Fkh only) fates (*Weigel et al., 1990*; *Casanova, 1990*), specifically by repressing the timer genes (both regions), activating Fkh (both regions), and differentially regulating Wg (repressed by Hkb in region 4). In region 1 (trunk), where Tll and Hkb are not expressed, gene expression is shaped by the intrinsic dynamics of the timer gene network: as D is activated and the level of Opa builds up, first Cad and then D are repressed. Finally, in region 2 (tail), these dynamics are modulated by transient expression of Tll, which delays the activation of D and Opa, and thereby prolongs the expression of Cad. Crucially, this Tll expression is weaker than in region 3, and so does not activate Fkh and (therefore) Wg.

### Recapitulation of mutant phenotypes

Simulated mutants of the 'outputs' Fkh, Cad, D, and Opa (*Figure 8D–G*) have perturbed gene expression within specific regions, but the overall spatial organisation of the tissue is unaffected. In the *fkh⁻* mutant, Wg is never activated in region 3. In the *D⁻* and *opa⁻* mutants, the turnover of timer gene expression in region 1 is perturbed: the repression of Cad is delayed in *D⁻*, and the repression of D is delayed in *opa⁻*. Finally, in the *cad^(m-z-)* mutant, widespread effects on gene expression coexist with fairly normal spatial organisation: in regions 3 and 4, Fkh and (therefore) Wg are not expressed, while in regions 1 and 2 the activation of D is reduced. (Although we modelled mutants as deficiencies and therefore did not recapitulate the delayed *cad* repression seen in *cad^(m-z-)* embryos [*Figure 4A*], we can interpret this delay as a knock-on effect of the reduced D expression, since D represses Cad.)

In contrast, simulated mutants involving the 'inputs' Tll and Hkb (*tor⁻*, *hkb⁻*, *tll⁻*, *tll⁻ opa⁻*; *Figure 8H–K*) show more serious spatial effects, which tend to resemble homeotic transformations. The *tor⁻* mutant, which removes all expression of Tll and Hkb, transforms regions 2–4 into region 1. The *hkb⁻* mutant essentially transforms region 4 (posterior midgut) into region 3 (hindgut). The *tll⁻* mutant transforms region 2 into region 1 but produces novel expression dynamics in region 3: D expression is transiently repressed (as in wild-type region 2) but Opa is not, producing a posteriorly shifted, transient Cad stripe and precluding any late expression of D. Finally, in the *tll⁻ opa⁻* mutant, the repression from Opa on D and Cad seen in the *tll⁻* mutant is removed, and region 3 is fully transformed into region 2.

### Discrepancies with real embryos

The discrepancies with real patterning stem from the simple, qualitative nature of the model. The activation of `Fkh` and (therefore) `Wg` is spuriously delayed in the *hkb*⁻ simulation (**Figure 8I**), owing to the discrete implementations of time, `Tll` expression, and `Fkh` regulation. The model cannot recapitulate the subtle shifting dynamics with the tail region (**Figure 3**) because the tail is modelled as a single, discrete block. Similarly, the model cannot recapitulate the concerted fate map shifts seen in *hkb*⁻ and *D*⁻ mutants (**Figure 4B**; **Figure 6E–H**) because there is no representation of region size. That said, if we extrapolate from the existing results, we can interpret the posterior shifting dynamics within the tail region as resulting from the posterior retraction of Tll expression over time (**Figure 5—figure supplement 3**; **Figure 5—figure supplement 4**), interpret the posterior fate map shift in *hkb*⁻ mutants as resulting from (indirect) cross-activation of *tll* by Hkb (**Figure 6H**), and interpret the anterior fate map shift in *D*⁻ mutants as resulting from potential cross-repression of *tll* by D.

### Summary

The genetic interactions we uncovered in this study are able to explain the qualitative aspects of timer gene patterning in both wild-type and mutant genotypes. In particular, our model explains how a graded Tll domain delineates both the anterior and posterior boundaries of the tail region, and explains why transient expression of Tll within the tail region is important for producing its characteristic timer gene dynamics. The model also explains the posteriorly shifted tail-like expression domains seen in *tll*⁻ and *tll*⁻ *opa*⁻ mutants as the result of graded and dynamic Hkb expression. For insight into quantitative phenomena such as the fate map shifts in *hkb*⁻ and *D*⁻ mutants, it will be necessary to analyse quantitative models incorporating zygotic cross-regulation of *tll*.

## Discussion

In this study, we have used mutants, multiplexed imaging, and modelling to elucidate how the blastoderm expression dynamics of the *Drosophila* timer genes *cad*, *D*, and *opa* arise from a combination of cross-regulatory interactions and spatially localised inputs from the posterior terminal system. This work has four main implications. First, we have demonstrated that timer gene expression is partially driven by intrinsic network dynamics. Second, we have uncovered more evidence that the timer genes have broad effects on developmental timing, through our discovery that *cad*ᵐ⁻ᶻ⁻ embryos precociously express genes associated with neural differentiation. Third, we have produced a coherent model for the patterning of the posterior terminal region. Fourth, we have clarified the segmental nature of the *Drosophila* tail. These findings increase our understanding of *Drosophila* development and have evolutionary significance for the mechanisms of axial patterning in other species.

### Timer gene expression is regulated by intrinsic network dynamics and extrinsic spatiotemporal inputs

This work provides evidence for a set of cross-regulatory interactions between *cad*, *D*, and *opa* that helps generate dynamic, sequential expression. In particular, we find that Cad activates *D* (i.e., promotes the expression of the next gene in the sequence), while D represses *cad* and Opa represses *cad* and *D* (i.e., both inhibit the previous gene(s) in the sequence). *opa* is not cross-regulated, however, making it an 'input-only' component of the three gene network (at least in the blastoderm context).

Timer gene expression is also shaped by extrinsic spatiotemporal regulation. In this work, we show how the timer gene network interacts with the posterior terminal system: most notably, Tll differentially represses *cad*, *D* and *opa* in the tail region, indirectly allowing *cad* expression to be maintained. The localised inputs from the posterior terminal system are overlaid on global temporal regulation provided by the nuclear:cytoplasmic ratio (which is particularly important for regulating the onset of *opa* transcription; ***Lu et al., 2009***) as well as the levels of maternal factors such as Tramtrack (***Harrison and Travers, 1990***; ***Brown et al., 1991***; ***Read et al., 1992***), Zelda (***Liang et al., 2008***; ***Harrison et al., 2011***; ***Nien et al., 2011***; ***McDaniel et al., 2019***), Stat92e (***Yan et al., 1996***; ***Hou et al., 1996***; ***Tsurumi et al., 2011***), and GAGA Factor/Trithorax-like (***Farkas et al., 1994***; ***Bhat et al., 1996***; ***Moshe and Kaplan, 2017***; ***Gaskill et al., 2021***). Ironically, precisely because these maternal factors are so crucial to development, their patterning roles remain less well understood than those of the zygotic patterning genes, which are less pleiotropic and therefore easier to study.

## Timer gene expression has broad effects on developmental timing

Recent work in the *Drosophila* blastoderm has demonstrated the extensive effects of timer genes on developmental gene expression. Opa has been shown to act as a pioneer factor, reshaping gene expression genome-wide by opening chromatin at hundreds of target enhancers (*Soluri et al., 2020*; *Koromila et al., 2020*). Cad and D are also known to regulate expression across the genome (*Li et al., 2008*; *MacArthur et al., 2009*; *Aleksic et al., 2013*). Here, we have found that early Cad expression appears to be necessary for the correct timing of later developmental events because neuroectodermal gene expression turns on precociously in *cad*[m-z-] embryos. The vertebrate Cad ortholog Cdx4 has also been shown to temporally regulate neural differentiation, in the developing spinal cord (*Joshi et al., 2019*), a tissue in which D and Opa orthologs play key developmental roles (reviewed in *Graham et al., 2003*; *Merzdorf, 2007*; *Houtmeyers et al., 2013*; *Stevanovic et al., 2021*). More generally, comparative evidence suggests that Cad/Cdx plays a deeply conserved role in the formation of the posterior body and the patterning of the posterior gut (*Copf et al., 2004*; *Wu and Lengyel, 1998*; *van Rooijen et al., 2012*; *Zhong et al., 2020*). In this context, *Drosophila cad*[m-z-] mutants offer a rare opportunity to study the genome-wide effects of a total loss of Cad/Cdx function without also catastrophically perturbing early developmental events.

## A revised picture of posterior terminal patterning in *Drosophila*

In this work, we have investigated blastoderm gene expression downstream of the posterior terminal system, revisiting a patterning network that was most intensely studied in the late 1980s and early 1990s (*Strecker et al., 1986*; *Mahoney and Lengyel, 1987*; *Mlodzik and Gehring, 1987b*; *Strecker et al., 1988*; *Jürgens and Weigel, 1988*; *Weigel et al., 1990*; *Casanova, 1990*; *Brönner and Jäckle, 1991*; *Wu and Lengyel, 1998*). The modern availability of marked balancers and multiplexed imaging techniques has allowed us to clarify the topology and spatiotemporal dynamics of the network, and incorporate genes (*D* and *opa*) that had not been cloned at the time most of the original work was completed. All told, we have identified 11 new regulatory interactions involved in *Drosophila* AP patterning, put forward the first formalised model (to our knowledge) for the patterning of the tail, and provided a solid foundation for future quantitative analyses of this system.

Although simple, our model provides new insights into how the tail and hindgut regions are specified in the early embryo. Both regions, along with segment A8, have long been known to depend on Tll expression (*Strecker et al., 1986*; *Diaz et al., 1996*). *tll* alleles can be arranged into a coherent phenotypic series in which the most posterior structures within the Tll-dependent region are the most sensitive to *tll* perturbation and the most anterior structures are the least (*Strecker et al., 1986*; Diaz et al., 1996), suggesting that this part of the blastoderm fate map is patterned by a gradient of Tll activity (*Casanova, 1990*). However, it has not been clear at the network level how graded Tll activity would be transduced into a specific series of boundaries and domains.

We found that *tll* expression was strong and persistent within the hindgut region, but weaker and transient in the tail region, with the anterior border of the expression domain retracting posteriorly across nuclei over time. We additionally found that Tll effectively patterned both the anterior and posterior boundaries of the tail region by differentially repressing *D* and *opa* relative to *cad*. Crucially, *D* and *opa* were repressed even where Tll expression was transient and weak, but *cad* was not repressed (and *fkh* was not activated) unless Tll expression was stronger, helping explain the transition from tail fate to hindgut fate as Tll levels increase. Furthermore, the retraction of the Tll domain over time explains the posterior shifting dynamics we found for the timer genes within the tail region, which contrasts with the anterior shifting dynamics previously described for the pair-rule and gap genes (*Jaeger et al., 2004*; *Keränen et al., 2006*; *Surkova et al., 2008*; *Lim et al., 2018*).

We also discovered, to our surprise, that there is a concerted posterior fate map shift in *hkb*[-] embryos, apparently mediated by a reduction in the size of the *tll* domain. (A subtle anterior fate map shift additionally occurs in *D*[-] embryos, which might also be mediated by Tll.) Although further research is necessary to determine the mechanism by which Hkb cross-regulates *tll*, the phenotype implies that the size of the *tll* domain is not an unmediated response to terminal signalling. (Indeed, there are hints in the existing literature that *tll* and *hkb* may be zygotically cross-regulated by other AP patterning genes as well; see *Casanova et al., 1994*; *Greenwood and Struhl, 1997*; *de las Heras and Casanova, 2006*.) These findings may complicate the interpretation of recent studies that have

characterised the input:output relationships between terminal signalling and *tll* and *hkb* expression using optogenetics (*Johnson and Toettcher, 2019*; *Johnson et al., 2020*; *Keenan et al., 2020*).

## The segmental character of the *Drosophila* tail

The ancestral insect body plan has 11 true abdominal segments plus the periproct/telson, but this number has been reduced in many extant insect lineages (*Snodgrass, 1935*; *Demerec, 1950*; *Matsuda, 1976*; *Chapman et al., 2013*). In *Drosophila*, the most common view has been that the embryo makes 10 abdominal segments (i.e., 15 parasegment boundaries), with the anal pads located in PS15/A10 (*Turner and Mahowald, 1979*; *DiNardo et al., 1985*; *Sato and Denell, 1986*; *Perkins and Perrimon, 1991*; *Kuhn et al., 1992*; *Schmidt-Ott et al., 1994*). In particular, territories corresponding to A8, A9, and A10 are visible at the morphological level during embryogenesis (*Turner and Mahowald, 1979*), and surveys of *en*, *wg*, *hh,* and *slp* staining have found evidence for (at most) 15 parasegment boundaries (*DiNardo et al., 1985*; *Baker, 1987*; *Baker, 1988*; *Kuhn et al., 1992*; *Grossniklaus et al., 1992*; *Mohler and Vani, 1992*; *Tabata et al., 1992*; *Lee et al., 1992*; *Tashiro et al., 1993*; *Kuhn et al., 1995*). However, fate mapping experiments (*Jürgens, 1987*) and surveys of *gooseberry* expression (*Baumgartner et al., 1987*; *Gutjahr et al., 1993*) have suggested that the embryo makes 16 parasegment boundaries, with the anal pads located in PS16/A11. There is also some evidence for A11 from patterns of gene expression in adult genital discs (*Freeland and Kuhn, 1996*).

Given the small size of the tail region within the embryo, the fact that it is covered by amnioserosa during key stages of patterning, and the fact that it later undergoes complicated morphogenetic rearrangements and fusions that obscure its metameric nature, it is perhaps unsurprising that the number of *Drosophila* segments has not been unambiguously resolved. In this study, we present evidence for a vestigial 16th parasegment boundary in the embryo by identifying additional domains of *slp* and *wg* expression and reinterpreting previously described domains of *eve* and *en*. These observations suggest that the anal pads are located in PS16. (Whether the tissue between PSB16 and the anus should be classified as a true 11th abdominal segment or a non-segmental periproct/telson is beyond the scope of this article.) However, PSB16 appears extremely dorsoventrally restricted and may have little functional significance in the organism. As the number of abdominal segments varies across insects (*Matsuda, 1976*), the mechanistic basis of this evolutionary reduction would be interesting to study within a comparative developmental framework.

Our findings suggest that the *Drosophila* embryo sequentially patterns two parasegment boundaries after gastrulation, and that in both cases the new boundary is patterned by abutting stripes of *slp* and *eve*. In PS15 and PS16, the relative arrangement of *slp*, *eve*, *wg,* and *en* expressing cells is the same conserved pattern that is found at parasegment boundaries in the *Drosophila* trunk and throughout the arthropod phylum (reviewed in *Clark et al., 2019*). However, tail segmentation differs from trunk segmentation in that resolved, stable *eve* stripes emerge de novo and with single-segmental periodicity, rather than from a dynamic and double-segmental phase of pair-rule gene expression.

Intriguingly, a remarkably similar switch from double-segment to single-segment periodicity occurs towards the end of segmentation in the centipede *Strigamia maritima*, where stable, resolved *eve* stripes start appearing de novo in the anterior segmentation zone instead of emerging from posterior oscillatory expression (*Brena and Akam, 2013*). A possible switch from double-segmental to single-segmental patterning has also been reported for terminal segments in the beetle *Tribolium* (*Janssen, 2014*). These observations hint that terminal and trunk segments may be homonomous at the level of segment-polarity gene expression but derived from distinct ontogenetic programs. More work is needed to determine how such a developmental switch—if present—is controlled, as well as its relationship to the more general problem of terminating axial development.

## Comparative analysis and evolutionary implications

We end this study by assessing the relevance of our findings from *Drosophila* to the development of other insect species. Which aspects of the *Drosophila* network are likely to be conserved in other insect species that have been used to study segmentation, such as *Tribolium*, *Nasonia vitripennis*, and *Oncopeltus fasciatus*? And how might the *Drosophila* network differ from that of its sequentially segmenting ancestors?

The cross-regulatory interactions that we found between the timer genes might be quite widely conserved in insect segmentation. Activation of *D* by Cad, repression of *cad* by Opa, and repression

of *D* by Opa are all consistent with a segment addition zone that is subdivided into a posterior region that expresses Cad and D and an anterior region that expresses Opa, as seen, for example, in *Tribolium* (*Clark and Peel, 2018*). However, repression of *cad* by D would need to be reconciled with the sustained expression of both *cad* and *D* in the posterior segment addition zone. Intriguingly, some of the timer gene cross-regulatory interactions may even be important for regulating expression dynamics in completely different developmental contexts, given that Opa has recently been found to repress *D* during the temporal patterning of *Drosophila* intermediate neural progenitors (*Abdussel-amoglu et al., 2019*).

The different components of the *Drosophila* terminal system seem to have acquired their posterior patterning roles at different times: posterior *tll* expression is found across diverse holometabolan species (*Schroder et al., 2000*; *Lynch et al., 2006*; *Wilson and Dearden, 2009*; *García-Solache et al., 2010*; *Lemke et al., 2010*; *Klomp et al., 2015*) although not in hemipterans (*Weisbrod et al., 2013*; *Bickel et al., 2013*), whereas *hkb* and *tor* appear to have been recruited to terminal patterning roles more recently (*García-Solache et al., 2010*; *Kittelmann et al., 2013*; *Duncan et al., 2013*). In *Tribolium*, *tll* is expressed downstream of *tor* (as in *Drosophila*), and *tor* RNAi embryos fail to express *cad* and *wg* in the posterior of the embryo, resulting in AP truncation (*Schoppmeier and Schröder, 2005*). In *Nasonia*, *tll* RNAi results in a reduction of posterior *cad*, as well as in gap gene misregulation that disrupts much of abdominal segmentation (*Lynch et al., 2006*). It will be instructive to test whether these losses of *cad* expression in *Tribolium* and *Nasonia* are mediated by ectopic expression of Opa, as we found for *tll⁻* and *tor⁻* mutants in *Drosophila*. If so, it would suggest that the initial spatial regulation of the timer gene network by Tll in the posterior blastoderm might be conserved across holometabolan embryos, despite their varying modes of development.

So, how *does* timer gene regulation differ between sequentially segmenting embryos (which establish a persistent segment addition zone) and simultaneously segmenting embryos like *Drosophila*? One key difference is likely to be the role of a posterior Wnt signalling centre: there is evidence from many different sequentially segmenting species that Wnt signalling is important for activating *cad* expression and maintaining the segment addition zone (reviewed in *Clark et al., 2019*), whereas we found that timer gene expression was unaffected in *Drosophila wg⁻* mutants, at least during our stages of interest. In addition, it seems probable that timer gene cross-regulation of *opa* is important in sequentially segmenting species, with this having been lost from the *Drosophila* lineage during the evolution of simultaneous patterning.

If we modify the *Drosophila* timer gene network to incorporate these additional features (Appendix 4), we can see how appropriate segment addition zone dynamics might naturally emerge (*Figure 8—figure supplement 1*). It therefore seems plausible that the cross-regulatory interactions between the *Drosophila* timer genes may represent an evolutionary vestige of a 'dynamical module' that was originally involved in axial elongation (*Clark and Peel, 2018*; *Clark, 2021*). Functional experiments in sequentially segmenting species will be necessary to test this hypothesis.

# Materials and methods

## Key resources table

| Reagent type (species) or resource | Designation | Source or reference | Identifiers | Additional information |
|---|---|---|---|---|
| Gene (*Drosophila melanogaster*) | *caudal* (*cad*) | FlyBase | FLYB:FBgn0000251 | |
| Gene (*D. melanogaster*) | *Dichaete* (*D*) | FlyBase | FLYB:FBgn0000411 | |
| Gene (*D. melanogaster*) | *engrailed* (*en*) | FlyBase | FLYB:FBgn0000577 | |
| Gene (*D. melanogaster*) | *even-skipped* (*eve*) | FlyBase | FLYB:FBgn0000606 | |
| Gene (*D. melanogaster*) | *forkhead* (*fkh*) | FlyBase | FLYB:FBgn0000659 | |
| Gene (*D. melanogaster*) | *huckebein* (*hkb*) | FlyBase | FLYB:FBgn0261434 | |
| Gene (*D. melanogaster*) | *muscle segment homeobox* (*msh*) | FlyBase | FLYB:FBgn0000492 | |
| Gene (*D. melanogaster*) | *odd-paired* (*opa*) | FlyBase | FLYB:FBgn0003002 | |

*Continued on next page*

*Continued*

| Reagent type (species) or resource | Designation | Source or reference | Identifiers | Additional information |
|---|---|---|---|---|
| Gene (*D. melanogaster*) | *sloppy-paired* (*slp*) | FlyBase | FLYB:FBgn0003430 | |
| Gene (*D. melanogaster*) | *tailless* (*tll*) | FlyBase | FLYB:FBgn0003720 | |
| Gene (*D. melanogaster*) | *torso* (*tor*) | FlyBase | FLYB:FBgn0003733 | |
| Gene (*D. melanogaster*) | *wingless* (*wg*) | FlyBase | FLYB:FBgn0284084 | |
| Strain, strain background (*D. melanogaster*) | Oregon-R | Bloomington Drosophila Stock Center | BDSC:5; RRID:BDSC_5 | 'Wild-type' |
| Strain, strain background (*Escherichia coli*) | One Shot BL21 Star (DE3) | Thermo Fisher Scientific | C601003 | Chemically competent cells |
| Genetic reagent (*D. melanogaster*) | cad[3] | Bloomington Drosophila Stock Center | BDSC:5316; FLYB:FBal0001531; RRID:BDSC_5316 | Gift from H. Skaer |
| Genetic reagent (*D. melanogaster*) | cad[2] FRT40A | Bloomington Drosophila Stock Center | BDSC:7091; FLYB:FBal0001530; FLYB:FBti0002071; RRID:BDSC_7091 | |
| Genetic reagent (*D. melanogaster*) | D[r72] | Bloomington Drosophila Stock Center | BDSC:8858 FLYB:FBal0086878; RRID:BDSC_8858 | Gift from S. Russell |
| Genetic reagent (*D. melanogaster*) | fkh[6] | Bloomington Drosophila Stock Center | BDSC:545; FLYB:FBal0004012; RRID:BDSC_545 | Gift from K. Roeper |
| Genetic reagent (*D. melanogaster*) | hkb[A321R1] | Bloomington Drosophila Stock Center | BDSC:2059; FLYB:FBal0031495; RRID:BDSC_2059 | |
| Genetic reagent (*D. melanogaster*) | opa[8] | Bloomington Drosophila Stock Center | BDSC:5335; FLYB:FBal0013272; RRID:BDSC_5335 | |
| Genetic reagent (*D. melanogaster*) | Df(3R)Exel6217 | Bloomington Drosophila Stock Center | BDSC:7695; FLYB:FBab0038272; RRID:BDSC_7695 | Deficiency covering the *tll* locus |
| Genetic reagent (*D. melanogaster*) | tor[XR1] | *Sprenger et al., 1989* | FLYB:FBal0016988 | Gift from T. Johnson |
| Genetic reagent (*D. melanogaster*) | wg[I-8] | Bloomington Drosophila Stock Center | BDSC:5351; FLYB:FBal0018500; RRID:BDSC_5351 | |
| Genetic reagent (*D. melanogaster*) | CyO, hb-lacZ | Bloomington Drosophila Stock Center | BDSC:6650; FLYB:FBba0000025; FLYB:FBti0002621; RRID:BDSC_6650 | |
| Genetic reagent (*D. melanogaster*) | TM6C, twi-lacZ | Bloomington Drosophila Stock Center | BDSC:7251; FLYB:FBba0000071; FLYB:FBti0010595; RRID:BDSC_7251 | |
| Genetic reagent (*D. melanogaster*) | TM3, hb-lacZ | Bloomington Drosophila Stock Center | BDSC:78357; FLYB:FBba0000047; FLYB:FBti0010581; RRID:BDSC_78357 | Gift from S. Russell |
| Genetic reagent (*D. melanogaster*) | hsFLP | Bloomington Drosophila Stock Center | BDSC:6; FLYB:FBti0002044; RRID:BDSC_6 | |
| Genetic reagent (*D. melanogaster*) | ovoD1 FRT40A | Bloomington Drosophila Stock Center | BDSC:2121; FLYB:FBtp0000359; FLYB:FBti0002071; RRID:BDSC_2121 | No longer listed in BDSC |
| Antibody | Anti-D (rabbit polyclonal) | *Soriano and Russell, 1998* | | (1:10) |
| Antibody | Anti-Hkb (rat polyclonal) | *Ashyraliyev et al., 2009* | | (1:100) |
| Antibody | Anti-Opa (guinea-pig polyclonal) | This paper | | (1:5000) |
| Antibody | Anti-Tll (rabbit polyclonal) | *Kosman et al., 1998* | | (1:100) |
| Antibody | Anti-guinea pig Alexa Fluor 647 (goat polyclonal) | Invitrogen | Cat#:A-21450; RRID:AB_2735091 | (1:1000) |
| Antibody | Anti-rabbit Alexa Fluor 488 (goat polyclonal) | Invitrogen | Cat#:A-11034; RRID:AB_2576217 | (1:1000) |
| Antibody | Anti-rabbit Alexa Fluor 555 (goat polyclonal) | Invitrogen | Cat#:A-21429; RRID:AB_2535850 | (1:1000) |
| Antibody | Anti-rat Alexa Fluor 488 (goat polyclonal) | Invitrogen | Cat#:A-11006; RRID:AB_2534074 | (1:1000) |

*Continued*

| Reagent type (species) or resource | Designation | Source or reference | Identifiers | Additional information |
|---|---|---|---|---|
| Recombinant DNA reagent | FI01113 (clone) | Drosophila Genomics Resource Center | DGRC:1623347; RRID:DGRC_1623347 | *opa* cDNA |
| Recombinant DNA reagent | Gateway pDONR221 (plasmid) | Thermo Fisher Scientific | Cat#:12536017 | |
| Recombinant DNA reagent | Gateway pET-DEST42 (plasmid) | Thermo Fisher Scientific | Cat#:12276010 | |
| Sequence-based reagent | *cad* | Molecular Instruments | HCR v3.0 probes | Designed to target NCBI:NM_134301.4 |
| Sequence-based reagent | *cad*-Intron | Molecular Instruments | HCR v3.0 probes | Designed to target NCBI:NT_033779.5: 20771910–20781798 |
| Sequence-based reagent | *D* | Molecular Instruments | HCR v3.0 probes | Designed to target NCBI:NM_001274901.1 |
| Sequence-based reagent | *en* | Molecular Instruments | HCR v3.0 probes | Designed to target NCBI:NM_078976.4 |
| Sequence-based reagent | *eve* | Molecular Instruments | HCR v3.0 probes | Designed to target NCBI:NM_078946.4 |
| Sequence-based reagent | *fkh* | Molecular Instruments | HCR v3.0 probes | Designed to target NCBI:NM_001300645.1 |
| Sequence-based reagent | *hkb* | Molecular Instruments | HCR v3.0 probes | Designed to target NCBI:NM_079497.4 |
| Sequence-based reagent | *msh* | Molecular Instruments | HCR v3.0 probes | Designed to target NCBI:NM_057976.3 |
| Sequence-based reagent | *opa* | Molecular Instruments | HCR v3.0 probes | Designed to target NCBI:NM_079504.4 |
| Sequence-based reagent | *slp* | Molecular Instruments | HCR v3.0 probes | Designed to target NCBI:NM_057382.3 |
| Sequence-based reagent | *tll* | Molecular Instruments | HCR v3.0 probes | Designed to target NCBI:NM_079857.4 |
| Sequence-based reagent | *wg* | Molecular Instruments | HCR v3.0 probes | Designed to target NCBI:NM_078778.5 |
| Sequence-based reagent | *lacZ* | Molecular Instruments | HCR v3.0 probes | Designed to target NCBI:NC_000913.3: c366305-363231 |
| Sequence-based reagent | B1-5 Alexa Fluor 488 | Molecular Instruments | HCR amplifiers | Amplifiers coordinated with probes |
| Sequence-based reagent | B1-5 Alexa Fluor 514 | Molecular Instruments | HCR amplifiers | Amplifiers coordinated with probes |
| Sequence-based reagent | B1-5 Alexa Fluor 546 | Molecular Instruments | HCR amplifiers | Amplifiers coordinated with probes |
| Sequence-based reagent | B1-5 Alexa Fluor 594 | Molecular Instruments | HCR amplifiers | Amplifiers coordinated with probes |
| Sequence-based reagent | B1-5 Alexa Fluor 647 | Molecular Instruments | HCR amplifiers | Amplifiers coordinated with probes |
| Sequence-based reagent | opaDM-F | This paper | PCR primers | AAAAAGCAGGCTTCGAAGGA GATAGAACCATGAACGCCTTCATTGAGC |
| Sequence-based reagent | opaA-R | This paper | PCR primers | AGAAAGCTGGGTTGTCGTAG CCGTGGGATG |
| Sequence-based reagent | attB1adap-F | This paper | PCR primers | GGGGACAAGTTTGTACAAAA AAGCAGGCT |
| Sequence-based reagent | attB2adap-R | This paper | PCR primers | GGGGACCACTTTGTACAAGA AAGCTGGGT |
| Commercial assay or kit | Gatteway BP Clonase II | Thermo Fisher Scientific | Cat#:11789020 | |
| Commercial assay or kit | Gateway LR Clonase II | Thermo Fisher Scientific | Cat#:11791020 | |
| Peptide, recombinant protein | Phusion Plus DNA Polymerase | Thermo Fisher Scientific | Cat#:F630S | |
| Chemical compound, drug | Overnight Express Instant TB Medium | Novagen | Cat#:71491-3 | |
| Chemical compound, drug | Ni-NTA Agarose | QIAGEN | Cat#:30210 | |
| Other | Normal Goat Serum blocking solution | Vector Laboratories | Cat#:S-1000-20 | |
| Other | DAPI stain | Invitrogen Scientific | Cat#:D1306 | (1 ng/µL) |
| Other | #1.5 coverslips | Corning | Cat#:2980-224 | |
| Other | SlowFade Gold AntiFade Mountant | Thermo Fisher Scientific | Cat#:S36940 | |

## *Drosophila* husbandry and genetics

Stock maintenance and embryo fixation (20 min with 4% formaldehyde in PBS) was performed as described in *Sullivan et al., 2000*. 'Wild-type' flies were Oregon-R. The mutant alleles used were $wg^{I-8}$ (Bloomington #5351), $cad^3$ (gift from H. Skaer), $cad^2$ (Bloomington #7091), $D^{r72}$ (gift from S. Russell), $opa^8$ (Bloomington #5340), $tor^{XR1}$ (gift from T. Johnson), $hkb^{A321R1}$ (Bloomington #2059), Df(3R)Exel6217 (Bloomington #7695, a deficiency covering the *tll* locus), and $fkh^6$ (gift from K. Roeper). Mutant lines obtained from the Bloomington *Drosophila* Stock Centre were verified by cuticle preparations as described in *Sullivan et al., 2000*. The $tll^-$ $opa^-$ double mutant was generated by the Cambridge Fly Facility by recombining Df(3R)Exel6217 and $opa^8$. Mutants were balanced over marked balancer chromosomes expressing *lacZ* during early embryogenesis: CyO, *hb-lacZ* (Bloomington #6650) for the second chromosome and TM6C, *twi-lacZ* $Sb^1Tb^1$ (Bloomington #7251) or TM3, *hb-lacZ* $Sb^1$ (gift from S. Russell) for the third.

   $cad^-$ germline clones were generated using the heatshock induced FLP/FRT system as described in *Selva and Stronach, 2007*. Briefly, eight vials of 30 $cad^2$ *FRT40A/CyO* virgin females (Bloomington #7091) were each crossed with 10 *hsFLP w; ovoD1 FRT40A/CyO* males (constructed by crossing Bloomington #6 *hsFLP w; Adv/CyO* females with Bloomington #2121 *ovoD1 FRT40A/CyO*, but note that #2121 is no longer listed in Bloomington). Adults were flipped to new vials every 2 days, resulting in a total of ~100 vials. When crawling L3 larvae were visible, vials were heatshocked at 37°C in a waterbath for 1 hr, allowed to recover at 25°C for 24 hr, then heatshocked again at 37°C for 1 hr. Approximately 600 non-*CyO* virgin females (some presumably with $cad^2/cad^2$ ovaries) were collected from the heatshocked vials and crossed with ~300 $cad^3$/*CyO, hb-lacZ* males. Resulting embryos without *lacZ* expression lacked both maternal and zygotic *cad* ($cad^{m-z-}$), while embryos with *lacZ* expression were paternal rescues ($cad^{m-z+}$). Zygotic *cad* mutants ($cad^{m+z-}$) were offspring from $cad^3$/*CyO, hb-lacZ* parents that lacked *lacZ* expression; note that this genotype is also heterozygous for maternal *cad*.

## Opa antibody generation

Clone FI01113 containing *opa* coding sequence was obtained from the Drosophila Genomics Resource Center. Gateway attB primers were designed to express 386 amino acids from the N-terminus of Opa (amino acids 3–389), spanning the zinc finger region in the centre of the protein. The forward primer included a Shine-Dalgarno sequence; the reverse primer was designed to be in-frame with the C-terminal fusion of the Gateway expression vector pET-DEST42 (Thermo Fisher Scientific). A two-stage PCR procedure was used to obtain a final amplicon carrying the attB-sequences at each end of the N-terminal *opa* sequence.

   Primers for the first amplification were
   paDM-F: AAAAAGCAGGCTTCGAAGGAGATAGAACCATGAACGCCTTCATTGAGC
   paA-R: AGAAAGCTGGGTTGTCGTAGCCGTGGGATG
   Overlapping primers for the second amplification to complete the attB regions were
   attB1adap-F: GGGGACAAGTTTGTACAAAAAAGCAGGCT
   attB2adap-R: GGGGACCACTTTGTACAAGAAAGCTGGGT

   The attB-opa amplicon was obtained by PCR with Phusion proofreading polymerase (Thermo Fisher Scientific) using primers opaDM-F and opaA-R. This first amplicon was diluted 1000-fold, then Phusion PCR was repeated with primers attB1adap-F and attB2adap-R. This attB-opa amplicon was recombined into Gateway donor vector pDONR (Thermo Fisher Scientific) using the BP Clonase II kit (Thermo Fisher Scientific). Plasmid DNA from a sequence-verified clone was then recombined into pET-DEST42 using the LR Clonase II kit (Thermo Fisher Scientific).

   For expression of the fusion protein, plasmid DNA was transformed into One Shot BL21 Star (DE3) chemically competent *Escherichia coli* (Thermo Fisher Scientific). Opa protein was expressed in two ways, firstly by IPTG induction of exponentially growing cells (0.75 mM IPTG for 2.75 hr), secondly by overnight culture in TB Overnight Express (Novagen). The Opa fusion protein in pET-DEST42 had a C-terminal 6-His tag. Protein was purified from bacterial pellets, each from 100 ml of cells induced in IPTG or TB Overnight Express. Purification was carried out using Ni-NTA Agarose (QIAGEN), under 8 M urea denaturing conditions according to the manufacturer's protocol. Purified protein was dialysed against water, then concentrated using an Amicon Ultra-Ultracel 5 kDa centrifugal filter (Millipore). Antibodies were raised in two guinea pigs by Eurogentec. Aliquots are available from EC on request.

## HCR in situ hybridisation and antibody staining

Prior to staining, fixed embryos stored in methanol were put through a rehydration series of 5 min each at 75, 50, and 25% methanol in PBS + 0.1% Tween-20, then washed three times with PBS + 0.1% Tween-20.

HCR in situ hybridisation was performed using probes and hairpins produced by Molecular Instruments, following the protocol for whole-mount fruit fly embryos included in *Choi et al., 2016*, adapted for v3.0 probes as described in *Choi et al., 2018*, with the following changes. Treatment of fixed embryos with ethanol, xylene, and proteinase K was omitted. The percentage of dextran sulphate in the probe hybridisation and amplification buffers was reduced from 10% w/v to 5% w/v, to reduce viscosity and allow the embryos to settle more easily in the tube. A 20 min postfix step (4% formaldehyde in 5× SSC + 0.1% Tween-20) was added at the end of the protocol to stabilise the signal.

For antibody staining following HCR, embryos were incubated for 30 min in blocking solution (5% Normal Goat Serum [Vector Laboratories] in 5× SSC + 0.1% Triton X-100), at room temperature with rocking. Embryos were then incubated overnight in preabsorbed primary antibody diluted in blocking solution, at 4°C with rocking. Embryos were washed four times for 15 min in 5× SSC + 0.1% Triton X-100, at room temperature with rocking, then incubated for 30 min in blocking solution, at room temperature with rocking. Embryos were then incubated for 2 hr with fluorescently labelled secondary antibody diluted in blocking solution at room temperature with rocking. Embryos were washed four times for 15 min then one time for 30 min with 5× SSC + 0.1% Triton X-100 at room temperature with rocking. Antibody staining without prior HCR was performed as above with the exception that PBS was used instead of 5× SSC. Primary antibodies were guinea pig anti-Opa (this work) at 1:5000, rabbit anti-Dichaete (*Soriano and Russell, 1998*) at 1:10, rabbit anti-Tll (*Kosman et al., 1998*) at 1:100, and rat anti-Hkb (*Ashyraliyev et al., 2009*) at 1:100. Secondary antibodies were goat anti-guinea pig Alexa Fluor 647 (Invitrogen A-21450), goat anti-rabbit Alexa Fluor 488 (Invitrogen A-11034), goat anti-rabbit Alexa Fluor 555 (Invitrogen A-21429), and goat anti-rat Alexa Fluor 488 (Invitrogen A-11006), diluted 1:1 with 100% glycerol for storage and used at 1:500 (1:1000 overall).

Following HCR and/or antibody staining, embryos were incubated for 30 min with 1 ng/μL DAPI (Thermo Fisher Scientific) in 5× SSC + 0.1% Tween-20, at room temperature with rocking, then washed three times for 30 min in 5× SSC + 0.1% Tween-20, at room temperature with rocking. Prior to mounting, embryos were stored in 1.5 mL tubes in SlowFade Gold Antifade Mountant (Thermo Fisher Scientific).

## Microscopy

Embryos were mounted in SlowFade Gold Antifade Mountant (Thermo Fisher) on glass microscope slides (Thermo Scientific) with #1.5 coverslips (Corning). #1.5 coverslips were used as bridges to prevent embryos from being squashed. Clear nail varnish was used to seal the edges of the slide.

Microscopy was performed on an Olympus FV3000 confocal microscope at the Department of Zoology Imaging Facility (University of Cambridge). Acquired images were 12-bit, with a 1024 × 768 scan format and a 2 μs/pixel dwell time. Whole embryo images were acquired using an Olympus UPlanSApo 30 ×1.05 NA silicon immersion oil objective, a physical pixel size of 0.47 μm × 0.47 μm, and a z-stack step size of 1.5 μm. The close-ups in *Figure 1* and *Figure 3* were acquired using an Olympus UPlanSApo 60 × 1.3 NA silicon immersion oil objective, a physical pixel size of 0.21 μm × 0.21 μm, and a z-stack step size of 0.8 μm. Each z-stack was specified so as to span from just above the top surface of the focal embryo through to the middle of its yolk.

In each experiment, embryos had been stained for up to four transcripts and/or proteins of interest plus nuclei, generally using Alexa Fluor 488, Alexa Fluor 546, Alexa Fluor 594, Alexa Fluor 647, and DAPI. (For mutant experiments, a *lacZ* probe or a probe to a gene covered by a deficiency was additionally labelled with one of these same fluorophores, so that homozygous mutant embryos could be easily identified.) All imaging channels were acquired sequentially to minimise cross-talk. The laser lines and collection windows were: 405 laser and 443–472 nm window for DAPI; 488 laser and 500–536 nm window for Alexa Fluor 488; 561 laser and 566–584 nm window for Alexa Fluor 546 or Alexa Fluor 555; 594 laser and 610–631 nm window for Alexa Fluor 594; 640 laser and 663–713 nm window for Alexa Fluor 647. Alexa Fluor 514 (514 laser and 519–540 nm window) was used in place of Alexa Fluor 488 for a round of HCR experiments carried out when the 488 laser was awaiting repair. When necessary, a transmitted light channel was also collected to allow for embryo staging based on the progress of cellularisation.

## Image analysis and figure preparation

Embryo staging was based on Bownes stages (*Bownes, 1975*; *Campos-Ortega and Hartenstein, 1997*), with subdivision of particular stages into substages where necessary (details in Appendix 1). Fiji (*Schindelin et al., 2012*) was used for routine inspection of imaging data and certain image adjustments (details in Appendix 2). Image processing and analysis scripts were written in `Python 3` (https://www.python.org) using the libraries `NumPy` (*Harris et al., 2020*), `SciPy` (*Virtanen et al., 2020*), `scikit-image` (*van der Walt et al., 2014*), and `matplotlib` (*Hunter, 2007*); see Appendix 2 for details. Figures were assembled in Affinity Designer (Serif Europe). Embryo outlines were drawn manually in Affinity. Image look-up tables (LUTs) were either chosen from the 'ChrisLUTs' LUT package for ImageJ (Christophe Leterrier and Scott Harden; https://github.com/cleterrier/ChrisLUTs; 'Neuro-Cyto LUTs' update site in Fiji) or generated for custom colours using a macro provided by Nicolás De Francesco (https://github.com/ndefrancesco).

## Models and simulations

Models were implemented in Python using `NumPy` (*Harris et al., 2020*), and outputs were plotted using `matplotlib` (*Hunter, 2007*). See Appendix 4 for details.

## Acknowledgements

This project was made possible by Michael Akam, who provided laboratory space, resources, encouragement, and helpful feedback on the manuscript. We are grateful to Ken Siggens for generating the guinea pig anti-Opa antibody, and to Simon Collier at the Department of Genetics Fly Facility (University of Cambridge) for creating the *tll⁻ opa⁻* double mutant. We thank the Imaging Facility at the Department of Zoology (University of Cambridge) for confocal imaging support, and members of the *Drosophila* community for various fly lines and reagents. EC thanks Angela DePace for hosting him in her group while work on this project was ongoing. Stocks and materials obtained from the Bloomington Drosophila Stock Center (NIH P40OD018537) and the Drosophila Genomics Resource Center (NIH 2P40OD010949) were used in this study. Information from FlyBase (*Larkin et al., 2021*) was invaluable.

## Additional information

### Funding

| Funder | Grant reference number | Author |
|---|---|---|
| Biotechnology and Biological Sciences Research Council | Research Grant BB/P009336/1 | Erik Clark |
| Trinity College, University of Cambridge | Junior Research Fellowship | Erik Clark |
| European Molecular Biology Organization | Postdoctoral Fellowship ALTF 383-2018 | Erik Clark |
| Deutsche Forschungsgemeinschaft | Research Fellowship BE 6732/1-1 | Matthew A Benton |
| Isaac Newton Trust | Research Grant | Matthew A Benton |
| Department of Zoology, University of Cambridge | | Matthew A Benton |
| Wellcome Trust | PhD Studentship | Margherita Battistara |

The funders had no role in study design, data collection and interpretation, or the decision to submit the work for publication. For the purpose of Open Access, the authors have applied a CC BY public copyright license to any Author Accepted Manuscript version arising from this submission.

## Author contributions
Erik Clark, Conceptualization, Software, Supervision, Funding acquisition, Investigation, Visualization, Methodology, Writing – original draft, Writing – review and editing, Data curation; Margherita Battistara, Investigation, Writing – review and editing; Matthew A Benton, Conceptualization, Supervision, Funding acquisition, Investigation, Methodology, Writing – original draft, Project administration, Writing – review and editing, Data curation, Visualization

## Author ORCIDs
Erik Clark ⓘ http://orcid.org/0000-0002-5588-796X
Matthew A Benton ⓘ http://orcid.org/0000-0001-7953-0765

## Decision letter and Author response
Decision letter https://doi.org/10.7554/eLife.78902.sa1
Author response https://doi.org/10.7554/eLife.78902.sa2

---

# Additional files

## Supplementary files
• MDAR checklist

## Data availability
All necessary data are included in the main text, appendices, and supplementary information. The confocal imaging dataset on which this study is based is freely available to download from the BioImage Archive (http://www.ebi.ac.uk/bioimage-archive; *Ellenberg et al., 2018*; *Sarkans et al., 2018*) under accession number S-BIAD582. This 335 GB dataset contains multiplexed image stacks of more than 800 individual embryos, including 12 different genotypes and over 50 different genotype / gene product combinations. Image analysis code and a sample image are provided in *Appendix 2—figure 1—source data 1*. A list of the corresponding image file(s) within the BioImage Archive dataset for all figure panels within the main text, appendices, and supplementary information is provided in *Appendix 2—figure 1—source data 2*. Source Data files are provided for the expression traces in the main figures and figure supplements.

The following dataset was generated:

| Author(s) | Year | Dataset title | Dataset URL | Database and Identifier |
| --- | --- | --- | --- | --- |
| Clark E, Battistara M, Benton MA | 2022 | A timer gene network is spatially regulated by the terminal system in the *Drosophila* embryo | https://www.ebi.ac.uk/biostudies/bioimages/studies/S-BIAD582 | BioImage Archive, S-BIAD582 |

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

## Appendix 1

### Embryo staging and selection

Embryos younger than stage 5 were staged to a nuclear cycle based on their nuclear density (stage 4.1 = nuclear cycle 10; stage 4.2 = nuclear cycle 11; stage 4.3 = nuclear cycle 12; stage 4.4 = nuclear cycle 13), while embryos older than stage 5 were staged by the progress of morphogenesis, the presence of mitotic domains (*Foe, 1989*), and/or the appearance of terminal segment-polarity stripes. Stage 5 itself was divided into five substages, stage 5.1 to stage 5.5, which can be differentiated from one another on the basis of *wg* expression, *D* expression, *eve* expression, or the progress of cellularisation (*Appendix 1—figure 1*; for most stains, we relied on *wg* and/or *D* expression). *Appendix 1—table 1* describes our staging criteria and also notes how our stage 5 classification scheme maps onto the eight 'temporal equivalence' classes used in *Surkova et al., 2008* and the four 'phases' used in *Schroeder et al., 2011*.

Expression patterns in embryos of the same (sub)stage and genotype generally looked remarkably similar; each phenotype we describe was observed in multiple individual embryos (biological replicates) and was consistent across different stain combinations (experiments). Occasional obviously atypical embryos (e.g., very small, or with abnormal patterns of mitotic division) were identified by visual inspection and discarded from the dataset. Any repeat scans of a given embryo were also discarded from the dataset to avoid pseudoreplication and artefacts from photo-bleaching. Embryos with the same stain combination were generally sourced from a single experiment; stainings were only repeated and/or combined when this was necessary to improve the coverage of stages. Embryos of all orientations were examined when characterising mutant phenotypes, but only laterally oriented embryos were selected for figure preparation and quantitative analysis. In most figures, a single representative embryo and/or expression trace is shown for any given (sub)stage. In some figures (*Figure 4*; *Figure 6*; *Figure 7*; *Figure 4—figure supplement 1*; *Figure 4—figure supplement 2*; *Figure 5—figure supplement 3*; *Figure 6—figure supplement 1*; *Figure 7—figure supplement 1*; *Figure 7—figure supplement 2*), expression traces from 2 to 4 embryos of the same stage and genotype are overlaid on the same axes to show the qualitative consistency of each phenotype across individuals. These sets of embryos were manually selected to be close matches in stage and orientation, since both factors influence the shape of the resulting expression trace.

**Appendix 1—table 1.** Embryo staging and substaging criteria used in this work.

Bownes stages 4, 5, 8, and 11 (*Bownes, 1975*) are further divided into substages, as described, based on the expression patterns of *wg*, *en*, *D*, and *eve*, or the progression of cellularisation.
For each stage 5 substage, the corresponding 'temporal equivalence class(es)' (*Surkova et al., 2008*) or 'phase' (*Schroeder et al., 2011*) are also listed for comparison. Note that the subdivision of a continuous developmental process into discrete timeclasses is convenient for analysis but biologically somewhat arbitrary; there are no sharp boundaries between the substages we have defined.

| Stage | Criteria | Surkova | Schroeder |
|---|---|---|---|
| 1–3 | Same as *Bownes, 1975*. | | |
| 4.1 | Syncytial blastoderm nuclear cycle 10 (judged by number/density of nuclei). | | |
| 4.2 | Syncytial blastoderm nuclear cycle 11 (judged by number/density of nuclei). | | |
| 4.3 | Syncytial blastoderm nuclear cycle 12 (judged by number/density of nuclei). | | |
| 4.4 | Syncytial blastoderm nuclear cycle 13 (judged by number/density of nuclei). | | |
| 5.1 | *wg*: not present.<br>*D*: broad trunk domain, head domain not well established (mainly nuclear dots).<br>*eve*: broad trunk expression, may have some AP modulation.<br>Blastoderm morphology: round, early-looking nuclei. | T1 | Phase 1 |
| 5.2 | *wg*: head and posterior domain expression just starting (mainly nuclear dots).<br>*D*: head domain established, trunk domain still uniform.<br>*eve*: not yet a regular 7 stripe pattern.<br>Blastoderm morphology: no invagination of plasma membrane. | T2–3 | Phase 2 |

*Appendix 1—table 1 Continued on next page*

*Appendix 1—table 1 Continued*

| Stage | Criteria | Surkova | Schroeder |
|---|---|---|---|
| 5.3 | *wg*: head and posterior domains established; wg0 and wg1 forming.<br>*D*: broad trunk domain becoming fainter in the middle. No tail domain.<br>*eve*: regular 7 stripe pattern but stripes still fuzzy and broad.<br>Blastoderm morphology: plasma membrane invaginating. | T4–5 | Phase 2 |
| 5.4 | *wg*: wg0 and wg1 well established; trunk stripes (mainly odd-numbered) just appearing.<br>*D*: tail domain appearing laterally with similar intensity to trunk expression; anterior and posterior halves of the trunk domain well-separated; trunk expression becoming more modulated.<br>*eve*: 7 well separated but still symmetrical stripes.<br>Blastoderm morphology: membranes have reached the bottom of the nuclei. | T6–7 | Phase 3 |
| 5.5 | *wg*: segmental pattern clearly developing (both odd-numbered and even-numbered stripes), though may not be fully established.<br>*D*: tail domain more established and separated from the trunk; trunk expression starting to fade; neuroectoderm expression just appearing, including a bright anterior-ventral stripe.<br>*eve*: anterior stripes narrowing to two-cell wide late element expression; posterior stripes becoming AP graded as they transition to the late element.<br>Blastoderm morphology: elongated nuclei. | T8 | Phase 4 |
| 6 | *wg*: regular segmental stripes.<br>*D*: tail domain strong; trunk expression (except dorsal saddle) fading; neuroectoderm expression developing but not yet uniform across the AP axis.<br>*eve*: all 7 stripes have narrowed, faint secondary stripes present.<br>Blastoderm morphology: signs of gastrulation and/or cephalic furrow formation; by late stage 6 pole cells moving dorsally and dorsal crumpling present. | | |
| 7 | Same as *Bownes, 1975*. | | |
| 8.1 | Mitotic domain 4 dividing. | | |
| 8.2 | wg14 absent/weak, en15 absent. | | |
| 8.3 | wg14 present, en15 absent/weak. | | |
| 8.4 | wg14 present, en15 present. | | |
| 9 | Same as *Bownes, 1975*. | | |
| 10 | Same as *Bownes, 1975*. | | |
| 11.1 | wg15 present, en16 absent. | | |
| 11.2 | wg15 present, en16 present. | | |
| 12+ | Same as *Bownes, 1975*. | | |

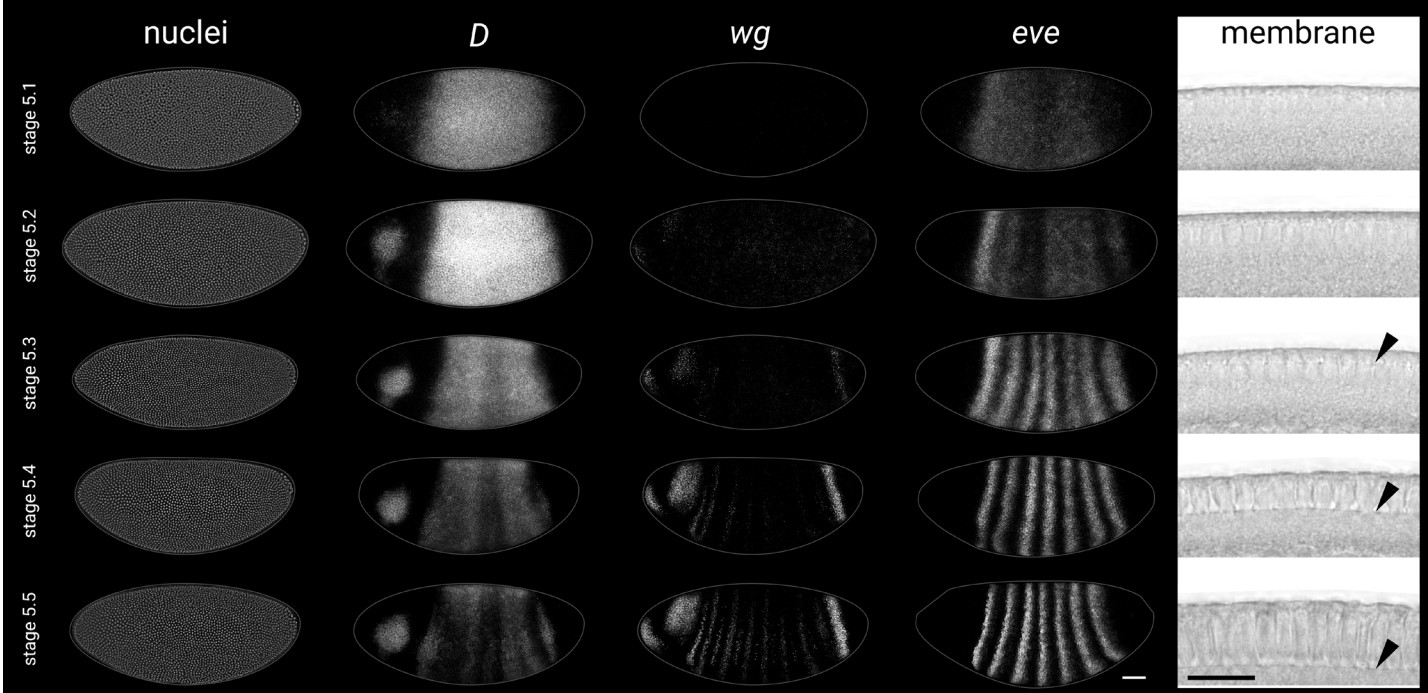

**Appendix 1—figure 1.** Substaging scheme for stage 5 embryos. Representative gene expression patterns and blastoderm morphology for each of the five substages in our substaging scheme for stage 5 (nuclear cycle 14). See *Appendix 1—table 1* for details. Rightmost column shows transmitted light images of a sagittal view of the dorsal blastoderm surface; black arrowheads point to the invaginating plasma membrane. All embryos are anterior left, dorsal up, lateral view. Scale bars = 50 µm (whole embryos), 20 µm (membrane close-ups); grey lines show embryo outlines. Note that the five images in each row are not all sourced from the same embryo.

## Appendix 2

### Initial image processing

Initial processing of raw image stacks was carried out to detect, rotate, mask, and crop each focal embryo (*Appendix 2—figure 1A–E*; *Appendix 2—figure 1—source data 1*, script 1). Briefly, a 'height map' of a given z-stack was built up by thresholding a maximum projection of z range 0:*i* for increasing values of *i*, and summing these together to produce an image showing the topography of any embryos within the field of view (*Appendix 2—figure 1B*). Local peaks within a truncated version of this image were detected and then used as the seeds for a watershed segmentation to separate touching embryos, while 'low-lying' background areas were masked (Appendix 2—figure 1C). Because all images were centred on a specific embryo of interest, the convex hull of the central segmented region was used as the embryo mask (*Appendix 2—figure 1D*). The major axis of the embryo mask was used to determine the orientation of this focal embryo, and the image was rotated accordingly so as to align its AP axis with the horizontal. The embryo mask was then dilated slightly, before being used to crop the image and mask non-embryo background (*Appendix 2—figure 1E*). This process was first applied to all images in batch, and then the resulting masks were inspected for accuracy. Any images with unsatisfactory masks were reprocessed individually, with manual parameter adjustment at the image segmentation step to correct the mask. Processed images were then flipped horizontally and/or vertically as necessary, to yield a consistent 'anterior left, dorsal up' orientation.

### Extraction of quantitative expression traces

Laterally oriented embryos of the appropriate stages and genotypes were then selected for the extraction of quantitative AP expression traces (*Appendix 2—figure 1F–L*; *Appendix 2—figure 1—source data 1*, scripts 2 and 3). Previous studies have tended to use percentage egg length to quantify AP expression profiles (e.g., *Pignoni et al., 1990*; *Surkova et al., 2008*; *Janssens et al., 2013*), but percentage egg length is not a perfect proxy for the AP axis due to the embryo's curvilinear intrinsic coordinate system (*Spirov et al., 2000*; *Luengo Hendriks et al., 2006*; *Spirov et al., 2013*). Percentage egg length measurements for expression domains near the poles are also potentially unreliable because they depend on the degree of flattening of a mounted embryo, given that a z-projection of a squashed embryo will exaggerate the size of the termini compared to a z-projection of an unsquashed embryo, due to the different curvature in z. We therefore decided to use a heuristic approach to approximate a curved trace along the lateral surface of the embryo, using guidance from embryo morphology.

Briefly, DAPI (nuclei)-derived height maps were used to define thin embryo 'shells' (~18 μm thick), which tracked the blastoderm surface in 3D and contained most of the gene expression signal (*Appendix 2—figure 1F and G*). A mean z-projection of the voxels within this shell region was then saved as a multichannel 2D image (*Appendix 2—figure 1H*). Next, the dorsal and ventral borders of the embryo mask were used to create a 'DV map' for this image by interpolation (*Appendix 2—figure 1I*), and 30% of the DV axis, corresponding to the mid-lateral part of the embryo, was selected for quantification (*Appendix 2—figure 1J*). As each embryo had a slightly different DV orientation on the slide, the selected DV range had to be adjusted manually for each image, so that the centre of the selected region consistently intersected with a DV position corresponding to the centre of the *D* head domain. This DV adjustment was important because the positions and expression intensities of most AP expression domains vary along the DV axis (*Keränen et al., 2006*). A 3D spline was fitted along the middle of the DV region of interest, using z values from the height map. To improve the consistency of the traces, the posterior endpoint of the spline was anchored close to a pixel coordinate marking the transition between the posterior midgut primordium and the pole cells, which was selected manually for each image. Cumulative distance along the spline was calculated in 3D using the Pythagorean theorem, accounting for the anisotropy of the z axis relative to the x and y. The total AP distance along the spline was normalised to 1, where 0 = the anterior tip of the embryo mask, and 1 = the beginning of the pole cells. Expression intensity traces were extracted for each channel by running a sliding window of 1% AP length (roughly 1 nuclear diameter) along the spline, with each window angled normal to the xy orientation of the spline (so as to avoid generating artificial expression overlaps from the slanted posterior domains), and bounded dorsally and ventrally by the DV region of interest (*Appendix 2—figure 1K*). Each extracted trace (*Appendix 2—figure 1J*) consisted of 500 measurements separated by a distance of 0.2% AP length.

The expression intensity traces in *Figure 3* (solid plotted lines) were calculated by moving a sliding window with a width of 25 pixels (~1 nuclear diameter) across the x axis of the rectangular region of interest and measuring the average intensity at 1 pixel intervals. Nuclear foci for *opa* and the *cad* intronic probe were identified by detecting local peaks above a threshold intensity; the dashed plotted lines in *Figure 3B* are density plots for the x coordinates of the detected foci.

## Normalised expression plots

When comparing traces from embryos of different stages to examine the dynamics of gene expression, all traces from a particular experimental sample were normalised to the range 0–1 as a group [i.e., for each channel, normalised values = (original values − min(group))/(max(group) − min(group))]. When comparing traces from individual embryos of the same stage to examine the positioning of expression domains within and between genotypes, each trace was normalised to the range 0–1 individually [i.e., normalised values = (original values - min(individual))/(max(individual) - min(individual))]. In $D^-$ mutants, expression levels were severely reduced across the entire AP axis, and so the normalised expression traces were multiplied by a small constant to dampen them. In most cases, expression traces are presented without any further adjustments. In *Figure 4—figure supplement 1B*, additional plots show 'aligned traces', in which each trace has been shifted anteriorly or posteriorly by a small amount so that the position of the anterior border of the *wg* posterior domain coincides in all traces. The aligned plots are useful for assessing any changes to the relative positioning of particular domains (as opposed to their absolute positional variation across different embryos).

## Embryo images

A list of source image files for all figure panels within the main text, appendices, and supplementary information is provided in *Appendix 2—figure 1—source data 2*. Unless otherwise stated, all embryo images shown in the display figures are maximum intensity projections of confocal z-stacks of the upper half of the embryo. Fiji was used to adjust image brightness and contrast, in accordance with guidelines presented by *Schmied and Jambor, 2020*. Image gamma was adjusted to 0.1 for all *opa* transcript stains, due to the extremely bright transcriptional foci. Embryos from the same round of staining and imaging are presented using the same brightness and contrast values; unless otherwise noted, this holds for any embryos within a given figure that share the same genotype and combination of stains. To correct for uneven illumination from the 405 laser, the DAPI (nuclei) signal from each blastoderm stage embryo was flattened by applying a Gaussian filter with σ = 6, and then dividing the original image by the new blurred image.

In *Figure 1B*, stage 8.1, the inset shows a maximum intensity projection from the surface to the midline of the embryo. In *Figure 1B*, stage 11.1, the inset shows a single section of a z-stack that was rotated −45° around the x axis using the ImageJ plug-in TransformJ (*Meijering et al., 2001*) using the 'Quintic B-Spline' method for interpolation. In *Figure 3*, the curved surface of the embryo was flattened in Fiji by reslicing each channel along the long axis of the embryo (output spacing 0.206 µm), manually masking the region of interest with a segmented line ('spline fit' checked) of width 130 pixels, using the 'Straighten' tool to process the entire stack, then reslicing the stack (output spacing 0.206 µm) and re-merging the channels to return to the original view. Average projections (DAPI [nuclei] and Opa channels) or maximum projections (HCR channels) were then generated for a z-range spanning from the top of the embryo to just below the nuclei.

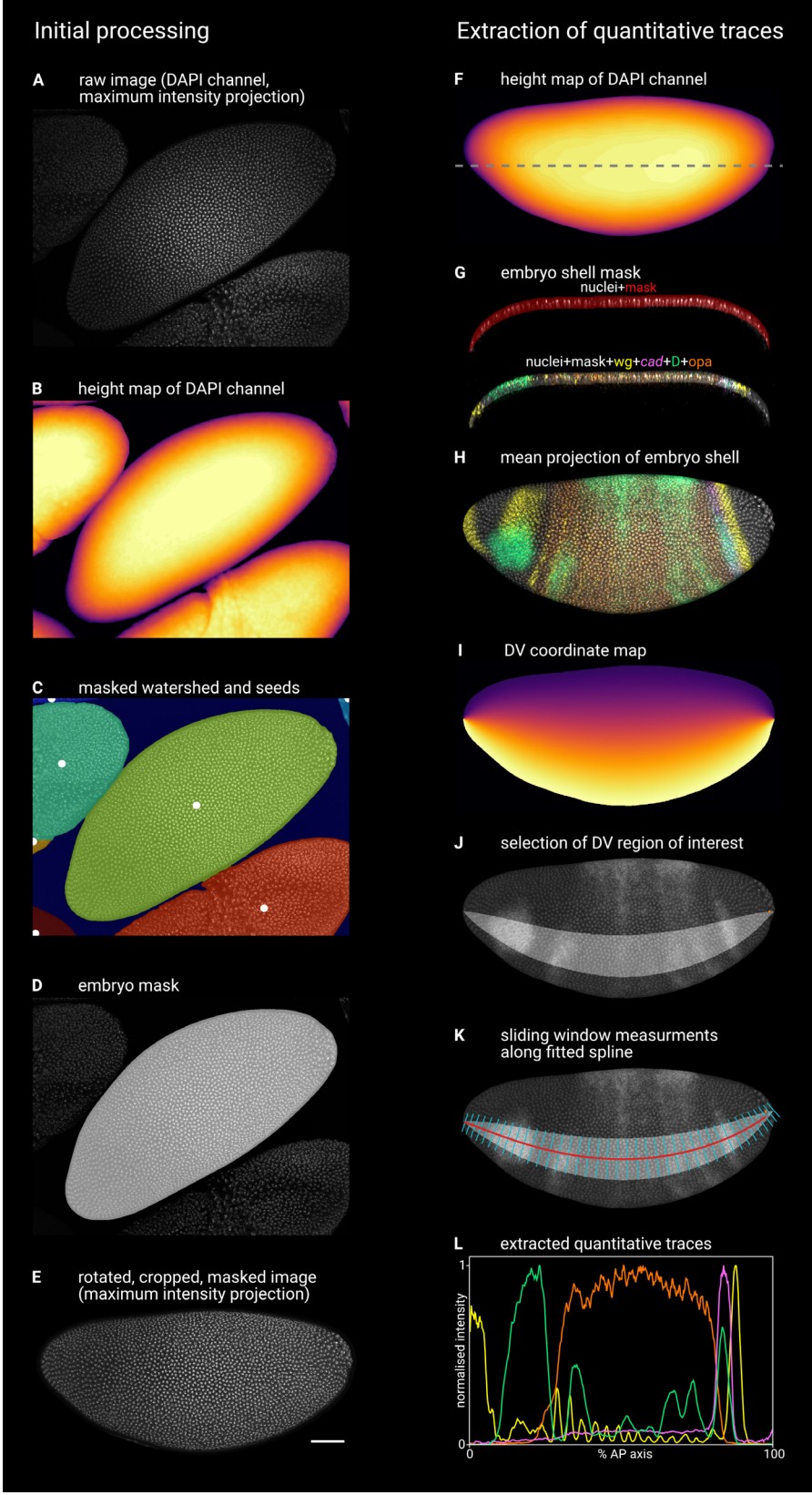

**Appendix 2—figure 1.** Illustration of image processing steps. Timer gene expression in timer gene mutants. Left column illustrates the initial processing of raw confocal data to generate single embryo stacks; right column illustrates the extraction of quantitative expression intensity traces from the processed stacks (see text for additional details and explanation). (**A**) Maximum z-projection of the DAPI (nuclei) channel of the raw multi-channel

stack. (**B**) Height map generated from the smoothed DAPI (nuclei) channel, where the colour of the pixel (yellow = high, black = low) indicates the height of the embryo surface. (**C**) Watershed segmentation of the (inverted) height map, in which local peaks (white dots) indicate the watershed seeds, each segmented region is overlaid by a different colour, and the low-lying background area is masked (dark blue). (**D**) The 2D mask for the focal embryo (light grey), overlaid on the maximum z-projection of the DAPI (nuclei) channel. (**E**) A maximum z-projection of the DAPI (nuclei) channel after image rotation, cropping, masking, and manual flipping. (**F**) Height map of the embryo z-stack, as in (**B**). Dashed line marks the location of the xz plane shown in (**G**). (**G**) An xz (frontal) section through the embryo, showing the embryo 'shell mask' overlaid on the DAPI (nuclei) channel (top) or on a merge of all five imaging channels (bottom). Note that the vast majority of the transcriptional signal is contained within the shell mask. (**H**) A 5-channel merge showing a mean z-projection of the 3D region defined by the embryo shell mask. (**I**) A map of the DV coordinates assigned to the embryo z-projection by interpolating between the dorsal and ventral boundaries of the embryo mask. (**J**) A region of interest (light grey area) defined by a specific range of DV coordinates. The range spans 30% of the DV axis and is selected to intersect with the round *D* domain in the head. Note the manually input coordinate (orange dot at the posterior of the embryo) marking the beginning of the pole cell region. (**K**) The red line shows a 2D projection of a 3D spline fitted to the centre of the DV region of interest (x and y coordinates) and the corresponding values from the height map (z coordinates). The cyan lines are normal to the spline in x and y. Expression traces are extracted from the image by running a sliding window (width = 1% of the length of the spline in 3D, anterior and posterior boundaries normal to the spline, dorsal and ventral boundaries defined by the DV region of interest) along the spline and recording the mean intensity of each image channel. (**L**) The quantitative expression traces extracted from the image, after normalising each trace to the 0–1 range.

The online version of this article includes the following source data for appendix 2—figure 1:

**Appendix 2—figure 1—source data 1.** Sample image stack and image analysis scripts.

**Appendix 2—figure 1—source data 2.** List of source image files for all figures.

# Appendix 3

## Justification for gene regulatory network topology

The detailed reasoning for the topology of the gene regulatory network in *Figure 8A* is presented in *Appendix 3—table 1*. This table summarises and discusses the experimental evidence relevant to each potential pairwise interaction between the genes in the network, drawing on the expression data from this study as well as a comprehensive survey of the existing literature on *Drosophila* posterior terminal patterning.

**Appendix 3—table 1.** Evidence for proposed cross-regulatory interactions between Tll, Hkb, Fkh, Wg, Cad, D, and Opa.

For every pairwise combination of input factor (Tll, Hkb, Fkh, Cad, D, or Opa) and potential target gene (*tll*, *hkb*, *fkh*, *wg*, *cad*, *D*, or *opa*), the inferred regulatory interaction (activation/repression/none/undetermined) is listed, accompanied by a summary of the relevant experimental evidence and lines of reasoning.

| Input | Target | Interaction | Evidence and discussion |
|---|---|---|---|
| Tll | *tll* | None | Tll is a dedicated repressor (***Morán and Jiménez, 2006***) so is unlikely to autoactivate, and sustained expression in wild-type (***Figure 5—figure supplement 1***) precludes strong autorepression. Indirect activation is a possibility, but one would need to look at a *tll*⁻ allele that still makes transcript to assess whether *tll* transcription is affected in *tll*⁻ mutants. |
| Tll | *hkb* | None | *hkb* is transcribed within the Tll domain (***Figure 5—figure supplement 1***), therefore Tll does not repress *hkb*. The *hkb* expression domain is a similar size in wild-type and *tll*⁻ embryos (***Figure 7B***), therefore Tll is not required to (indirectly) activate *hkb*. |
| Tll | *fkh* | (Indirect) activation | *fkh* is transcribed across the Tll domain in wild-type and *hkb*⁻ embryos (***Figure 7A and B***; ***Figure 7—figure supplement 1A***), and the *fkh* domain is reduced (to the size of the *hkb* domain) in *tll*⁻ embryos (***Figure 7A and B***; ***Figure 7—figure supplement 1B***). Activation from Tll is presumed to be indirect as Tll is a dedicated repressor (***Morán and Jiménez, 2006***). |
| Tll | *wg* | None | *wg* is transcribed within the Tll domain in wild-type embryos (***Figure 5B***) and in *hkb*⁻ mutants (***Figure 7A and B***; ***Figure 7—figure supplement 1A***), therefore Tll does not repress *wg*. Tll is necessary for *wg* expression (*wg* expression is lost in *tll*⁻ mutants and is posteriorly shifted in *hkb*⁻ mutants, correlating with the altered Tll domain; ***Figure 7A and B***; ***Figure 7—figure supplement 1A***), but this activation seems to be indirect (via Fkh) as Tll is a dedicated repressor (***Morán and Jiménez, 2006***), and the presence of Tll-positive, Hkb-negative territory is not sufficient to activate *wg* in *fkh*⁻ or *cad*^{m-z-} genotypes (***Figure 7***; ***Figure 7—figure supplement 1B***). |
| Tll | *cad* | (Weak) repression | *cad* transcription overlaps the graded anterior edge of the Tll domain throughout most of the blastoderm stage in wild-type embryos (***Figure 5***), indicating that Tll does not strongly repress *cad*. However, *cad* is still repressed in (Tll-positive, Fkh-positive) posterior tissue in *hkb*⁻ mutants (***Figure 6E–G***), suggesting that *cad* must be repressed by either Tll or Fkh (or both). As *cad* expression is largely normal in *fkh*⁻ mutants (***Figure 7C and D***), it seems likely that Tll does indeed repress *cad*, albeit more weakly than Tll represses other targets such as *D* and *opa*. Investigation of *cad* expression in *fkh*⁻ *hkb*⁻ double mutants would be informative for isolating the role of Tll in *cad* regulation. |
| Tll | *D* | Repression | The graded posterior border of the *D* domain is anticorrelated with Tll levels in wild-type embryos, and the *D* tail domain appears only after *tll* expression in this region decline (***Figure 5A***). The *D* posterior boundary shifts posteriorly in *tll*⁻ mutants, and also in *hkb*⁻ mutants, apparently because the *tll* domain is reduced (***Figure 6C–H***). *D* expression is normal in *fkh*⁻ mutants (***Figure 7C and D***), indicating that the repressive effect of Tll is not mediated by Fkh. It is currently unclear whether the *D* tail domain has the same regulatory logic / sensitivity to Tll as does the *D* trunk domain; investigation of *D* enhancer regions will be informative. |

*Appendix 3—table 1 Continued on next page*

*Appendix 3—table 1 Continued*

| Input | Target | Interaction | Evidence and discussion |
|---|---|---|---|
| Tll | opa | Repression | *opa* is excluded from the Tll domain in wild-type (**Figure 5A**), and the posteriorly shifting *opa* boundary (**Figure 3B**) correlates with the posteriorly shifting dynamics of the Tll domain (**Figure 5—figure supplement 3**; **Figure 5—figure supplement 4**). The *opa* boundary is shifted posteriorly in *tll*⁻ mutants, and also in *hkb*⁻ mutants, apparently because the *tll* domain is reduced (**Figure 6C–H**). *opa* expression is normal in *fkh*⁻ mutants (**Figure 7C and D**), indicating that the repressive effect of Tll is not mediated by Fkh. |
| Hkb | tll | (Indirect) activation | *tll* is coexpressed with Hkb (**Figure 5—figure supplement 1**), therefore Hkb does not repress *tll*. The *tll* domain is reduced in *hkb*⁻ mutants (**Figure 6F and H**), but as Hkb is a repressor (**Goldstein et al., 1999**) and *tll* expression is affected anterior to the Hkb domain, this *hkb*-dependent activation of *tll* is presumably indirect. *tll* expression also persists longer in *hkb*⁻ mutants (**Figure 6—figure supplement 2**), but again the Hkb-dependent effect on *tll* (in this case, late repression) is presumably indirect. |
| Hkb | hkb | None | Hkb is a repressor (**Goldstein et al., 1999**) so is unlikely to autoactivate. Sustained *hkb* expression in wild-type embryos (**Figure 5—figure supplement 1**) precludes strong autorepression. Indirect activation is a possibility, but one would need to look at a *hkb*⁻ allele that still makes transcript to assess whether *hkb* transcription is affected in *hkb*⁻ mutants. |
| Hkb | fkh | (Indirect) activation | *fkh* is transcribed across the Hkb domain (**Figure 7—figure supplement 1B**), therefore Hkb does not repress *fkh*. *fkh* is still expressed within the *hkb* domain in *tll*⁻ mutants (**Figure 7A and B**), indicating that Hkb can activate *fkh* independently of Tll. As Hkb is a repressor (**Goldstein et al., 1999**), this activation is presumably indirect. |
| Hkb | wg | Repression | The *wg* posterior boundary abuts the Hkb anterior boundary in wild-type embryos (**Figure 5B**), and *wg* expression extends to the posterior pole in *hkb*⁻ mutants (**Figure 6E and G**; **Figure 7A and B**). |
| Hkb | cad | Repression | *cad* is not expressed within the Hkb domain from early stage 5 in wild-type embryos (**Figure 5**; **Figure 5—figure supplement 2**), and clearance of *cad* expression from the posterior pole is delayed in *hkb*⁻ mutants (**Figure 6E**; **Figure 7A and B**). *cad* remains repressed from the Hkb domain in *tll*⁻ mutants (**Figure 6C, D and G**; **Figure 7A and B**), indicating that Hkb represses *cad* independently of Tll. In addition, *cad* expression is largely normal in *fkh*⁻ mutants throughout blastoderm stages (**Figure 7C and D**), suggesting that *cad* repression is not mediated by Fkh. Examining *cad* expression in *tll*⁻ *fkh*⁻ double mutants would be helpful to confirm whether Hkb represses *cad* independently of both Tll and Fkh. |
| Hkb | D | Repression | *D* expression is excluded from the Hkb domain in *tll*⁻ mutants (**Figure 6D**). Examining *D* expression in *tll*⁻ *fkh*⁻ double mutants would be helpful to confirm whether this repression is independent of Fkh. |
| Hkb | opa | Repression | *opa* expression is excluded from the Hkb domain in *tll*⁻ mutants (**Figure 6D**). Examining *opa* expression in *tll*⁻ *fkh*⁻ double mutants would be helpful to confirm whether this repression is independent of Fkh. |
| Fkh | tll | None | *tll* expression precedes Fkh expression in wild-type (**Figure 5—figure supplement 1**; **Weigel et al., 1989**; extended imaging dataset), therefore Fkh is not required to activate *tll*. Tll and Fkh are co-expressed throughout stage 5 (**Figure 7A and B**; extended imaging dataset), indicating that Fkh does not repress *tll*. In addition, although we did not examine *tll* expression in *fkh*⁻ mutants, *tll*-dependent patterning of *D* and *opa* appears normal in *fkh*⁻ mutants (**Figure 7C and D**), indicating that *tll* expression is unlikely to be perturbed. It would be useful to examine *tll* expression in *fkh*⁻ mutants to confirm this. |
| Fkh | hkb | None | *hkb* expression precedes Fkh expression in wild-type (**Figure 5—figure supplement 1**; **Weigel et al., 1989**; extended imaging dataset), therefore Fkh is not required to activate *hkb*. Hkb and Fkh are coexpressed throughout stage 5 (**Figure 7—figure supplement 1B**; extended imaging dataset), indicating that Fkh does not repress *hkb*. It would be useful to examine *hkb* expression in *fkh*⁻ mutants to confirm it looks normal. |

*Appendix 3—table 1 Continued on next page*

Appendix 3—table 1 Continued

| Input | Target | Interaction | Evidence and discussion |
|-------|--------|-------------|-------------------------|
| Fkh | *fkh* | None | Sustained Fkh expression in wild-type embryos (*Figure 7A*; *Weigel et al., 1989*; extended imaging dataset) suggests autorepression is unlikely. It would be useful to examine *fkh* expression in *fkh⁻* mutants to assess whether autoactivation occurs. |
| Fkh | *wg* | Activation | *wg* is only expressed in Fkh-positive, Hkb-negative territory in wild-type embryos (*Figure 5B*), and *wg* expression is strongly reduced in *fkh⁻* mutants (*Figure 7C and D*; *Figure 7—figure supplement 3*; *Figure 7—figure supplement 4*) and also *cad*^(m-z-) mutants (*Figure 4A and B*; *Figure 7A*; *Figure 4—figure supplement 3A*), which have reduced *fkh* expression (*Figure 7A and B*). |
| Fkh | *cad* | Undetermined | The posterior *cad* boundary consistently abuts the anterior *fkh* boundary, for example in wild-type embryos, *tll⁻* mutants, and *hkb⁻* mutants (*Figure 7A and B*). However, *cad* expression is largely normal in *fkh⁻* mutants throughout stage 5 (*Figure 7C and D*), with a possible posterior expansion after gastrulation (*Figure 7—figure supplement 3*). Because we think that Tll both represses *cad* and (indirectly) activates *fkh*, it is unclear whether Fkh indeed has no effect on *cad*, or alternatively whether Fkh and Tll repress *cad* redundantly. Misexpression of Fkh in the tail region would be informative. |
| Fkh | *D* | Undetermined | Unclear, as Fkh is only ever expressed in territories expressing *D* repressors Tll or Hkb (*Figure 7A and B*). Misexpression of Fkh in segmental territories would be informative. |
| Fkh | *opa* | Undetermined | Unclear, as Fkh is only ever expressed in territories expressing *opa* repressors Tll or Hkb (*Figure 7A and B*). Misexpression of Fkh in segmental territories would be informative. |
| Cad | *tll* | None | The *tll* domain emerges from Cad-positive territory (*Figure 5A*), therefore Cad does not repress *tll*. *tll* is expressed normally in *cad*^(m-z-) mutants (*Figure 7A and B*), therefore Cad is not required to activate *tll*. |
| Cad | *hkb* | None | The *hkb* domain emerges from Cad-positive territory (*Figure 2*; *Figure 5B*), therefore Cad does not repress *hkb*. *hkb* is expressed normally in *cad*^(m-z-) mutants (*Figure 7—figure supplement 1B*), therefore Cad is not required to activate *hkb*. |
| Cad | *fkh* | None | *fkh* expression is strongly reduced in *cad*^(m-z-) mutants, even though *tll* and *hkb* expression is largely normal (*Figure 7A and B*; *Figure 7—figure supplement 1*). |
| Cad | *wg* | Activation | Although the *wg* posterior domain is lost in *cad*^(m-z-) mutants (*Figure 4A and B*; *Figure 4—figure supplement 3A*), this phenotype is likely mediated by the loss of *fkh* expression in these embryos (*Figure 7A and B*), because the *wg* posterior domain is also lost in *fkh⁻* mutants (*Figure 7C and D*; *Figure 7—figure supplement 3*; *Figure 7—figure supplement 4*) and these have normal *cad* expression (*Figure 7C and D*). In addition, *wg* expression persists posterior to the *cad* domain throughout germband extension in wild-type, after broad blastoderm Cad expression has decayed (*Figure 2*). |
| Cad | *cad* | None | The persistent expression of *cad* in the tail in both wild-type embryos and *cad*^(m-z-) mutants (*Figure 4C*) is inconsistent with both direct autorepression and direct autoactivation. We interpret the delayed clearance of *cad* from the trunk in *cad*^(m-z-) mutants (*Figure 4C*) as due to the lower levels of *D* in this genotype (*Figure 4C*), rather than due to direct autoregulation. |
| Cad | *D* | Activation | *D* expression emerges within Cad-positive territory in both the trunk and the tail in wild-type embryos (*Figure 2*), indicating that Cad does not repress *D*. Reduced *D* levels in the trunk of *cad*^(m-z-) embryos (*Figure 4C*), dorsal loss of *D* tail expression in *cad*^(m-z-) embryos (*Figure 4*; *Figure 4—figure supplement 3A*), and late loss of *D* tail expression in *cad*^(m+z-) embryos (*Figure 4—figure supplement 3B*) all indicate that Cad activates *D*. However, additional activators of *D* must exist, given that *D* expression is reduced rather than completely lost in *cad*^(m-z-) embryos. |

Appendix 3—table 1 Continued on next page

*Appendix 3—table 1 Continued*

| Input | Target | Interaction | Evidence and discussion |
|---|---|---|---|
| Cad | *opa* | None | *opa* is transcribed strongly across the trunk while Cad levels are still high in wild-type embryos (**Figure 2**; **Figure 2—figure supplement 4**), and *opa* expression later invades the *cad* tail domain from the anterior (**Figure 3B**), indicating that Cad does not repress *opa*. *opa* is expressed largely normally in *cad*$^{m-z-}$ mutants (**Figure 4A**; **Figure 4—figure supplement 2**; though note the AP modulation), indicating that Cad is not required to activate *opa*. |
| D | *tll* | Undetermined | There is a subtle anterior shift and expansion of the posterior terminal fate map in *D*$^-$ mutants (**Figure 4A and B**; **Figure 4—figure supplement 2**), which can be most easily explained by supposing that D represses *tll*. Although we did not examine *tll* expression in *D*$^-$ mutants, *tll* and *D* are expressed in opposing gradients during stage 4.4 (nuclear cycle 13) in wild-type embryos (**Figure 5A**), and it seems plausible that mutual repression between *tll* and *D* could help to scale the AP pattern. Investigation of *tll* expression in *D*$^-$ mutants and misexpression of D in the posterior of the embryo would both be informative experiments. |
| D | *hkb* | Undetermined | Unclear, as domains of *D* and *hkb* expression are distinct (**Figure 5**). Misexpression of D in the posterior of the embryo would be informative. |
| D | *fkh* | Undetermined | Unclear, as domains of *D* and *fkh* expression are distinct (**Figure 2**; **Figure 7A and B**). Although we did not examine *fkh* expression in *D*$^-$ mutants, the *wg* posterior domain is activated normally in *D*$^-$ mutants (**Figure 4A and B**) suggesting that *fkh* expression is unlikely to be strongly affected. Misexpression of D in the posterior of the embryo would be informative. |
| D | *wg* | Undetermined | Unclear, as domains of *D* and *wg* expression are distinct (**Figure 2**). The *wg* posterior domain looks essentially normal in *D*$^-$ mutants (**Figure 4A and B**). Misexpression of D in the posterior of the embryo would be informative. |
| D | *cad* | Repression | *cad* expression ceases in the trunk as D levels increase in wild-type embryos (**Figure 2**; **Figure 2—figure supplement 3**; **Figure 2—figure supplement 4**), while *cad* expression persists in some parts of the trunk in *D*$^-$ mutants (**Figure 4A**). The anterior border of the *cad* tail domain correlates with the earlier position of the posterior border of the *D* trunk domain in wild-type, *hkb*$^-$ mutants, *tll*$^-$ mutants, and *tll*$^-$ *opa*$^-$ mutants (**Figure 2**; **Figure 6**; **Figure 4—figure supplement 5**; extended imaging dataset). |
| D | *D* | Undetermined | We were unable to assess possible autoregulatory effects, as *D* transcript levels were strongly reduced in the *D*$^-$ mutants we examined (**Figure 4A**), presumably due to nonsense-mediated decay. |
| D | *opa* | None | *opa* expression emerges from D-positive territory in the trunk in wild-type (**Figure 2**; **Figure 2—figure supplement 3**; **Figure 2—figure supplement 4**), and *opa* expression is largely normal in *D*$^-$ mutants (**Figure 4A**; **Figure 4—figure supplement 2**), indicating that D neither represses nor activates *opa*. The *opa* posterior border is shifted slightly anteriorly in *D*$^-$ mutants (**Figure 4—figure supplement 2**), but this is likely to be an indirect effect mediated by Tll, or possibly by Cad. |
| Opa | *tll* | Undetermined | Unclear, as the domains of *opa* and *tll* expression are distinct (**Figure 5A**). Misexpression of Opa in the posterior of the embryo would be informative. |
| Opa | *hkb* | Undetermined | Unclear, as domains of *opa* and *hkb* expression are distinct (**Figure 5**). Misexpression of Opa in the posterior of the embryo would be informative. |
| Opa | *fkh* | Undetermined | Unclear, as domains of *opa* and *fkh* expression are distinct (**Figure 5**; **Figure 7A and B**). Misexpression of Opa in the posterior of the embryo would be informative. |
| Opa | *wg* | Undetermined | Unclear, as domains of *opa* and (posterior) *wg* expression are distinct (**Figure 2**). Interestingly, Opa activates the segmental *wg* stripes in the trunk (**Benedyk et al., 1994**), but Opa/Zic is a Wnt antagonist in other developmental contexts (**Pourebrahim et al., 2011**; **Fujimi et al., 2012**; **Murgan et al., 2015**). Misexpression of Opa in the posterior of the embryo would be informative. |

*Appendix 3—table 1 Continued on next page*

*Appendix 3—table 1 Continued*

| Input | Target | Interaction | Evidence and discussion |
|-------|--------|-------------|-------------------------|
| Opa | *cad* | Repression | The anterior border of the *cad* tail domain retracts in wild-type embryos as Opa levels increase (**Figure 3B**), suggesting that Opa represses *cad*. Repression of *cad* by Opa is also suggested by the late repression of the ectopic *cad* expression present in the trunk of D⁻ mutants (**Figure 4A**; extended imaging dataset), and by the late repression of the *cad* posterior domain in *tll*⁻ mutants, which overlaps with *opa* expression (**Figure 6C and G**; **Figure 4—figure supplement 5**). |
| Opa | *D* | Repression | In wild-type embryos, *D* expression in the trunk decreases as Opa levels increase (**Figure 2**; **Figure 2—figure supplement 4**), and the anterior border of the *D* tail domain lines up with the Opa posterior border (**Figure 3A**). In *opa*⁻ mutants, *D* expression in the trunk persists for longer and *D* expression in the tail is strengthened (**Figure 4A and B**; **Figure 4—figure supplement 5**), indicating that Opa represses *D*. In addition, a *D* tail domain does not emerge in *tll*⁻ mutants, which misexpress *opa* anterior to the Hkb domain (**Figure 6C, D and G**), but a tail-like *D* domain does emerge in *tll*⁻ *opa*⁻ double mutants (**Figure 4—figure supplement 5**). |
| Opa | *opa* | None | Sustained *opa* expression in wild-type embryos (**Figure 2**; **Figure 2—figure supplement 4**) and normal expression of *opa* in *opa*⁻ mutants (**Figure 4A**; **Figure 4—figure supplement 2**; **Figure 4—figure supplement 5**) indicate that strong autoregulatory effects are unlikely, at least within our period of interest. |

## Appendix 4

### Model and simulation details

In *Figure 8*, the *Drosophila* AP axis is modelled as four discrete regions, where region 1 represents the trunk, region 2 represents the tail, region 3 represents the hindgut primordium, and region 4 represents the posterior midgut primordium. Each region receives hard-coded inputs from `Tll` and `Hkb`, logical variables that can take the values 0 (no expression), 1 (weak expression), or 2 (strong expression). Region 1 remains free of both `Tll` and `Hkb` expression across all four timepoints (0,0,0,0). Region 2 experiences weak, transient `Tll` expression (1,1,0,0) and no `Hkb` expression (0,0,0,0). Region 3 experiences rapidly-established strong `Tll` expression (1,2,2,2) and transient weak `Hkb` expression (1,1,0,0). Region 4 experiences rapidly-established strong `Tll` expression (1,2,2,2) and persistent strong `Hkb` expression (2,2,2,2).

In addition to `Tll` and `Hkb`, each region can express `Fkh`, `Wg`, `Cad`, `D`, and `Opa`, logical variables that can take one of either three (0, 1, 2) or two (0, 1) possible values, as defined by their regulatory logic:

`Fkh` = 1 if ((`Tll` + `Hkb`) > 1) and ((`Cad` + `Fkh`) > 0); else `Fkh` = 0.
`Wg` = 1 if (`Fkh` > 0) and (`Hkb` < 2); else `Wg` = 0.
`Cad` = 1 if (`D` < 2) and (`Opa` < 2) and (`Hkb` < 2) and (`Tll` < 2); else `Cad` = 0.
`D` = 2 if (`Opa` < 2) and (`Tll` < 1) and (`Hkb` < 1) and (`Cad` > 0); `D` = 1 if (`Opa` < 2) and (`Tll` < 1) and (`Hkb` < 1) and (`Cad` < 1); else `D` = 0.
`Opa` = 2 if (`Hkb` < 2) and (`Tll` < 1) and (`Opa` > 0); `Opa` = 1 if (`Hkb` < 2) and (`Tll` < 1) and (`Opa` < 1); else (`Opa` = 0).

Thus, `Fkh` is only expressed when combined `Tll` and `Hkb` levels are high, and `Cad` must initially be present for `Fkh` expression to become established. `Wg` is expressed when `Fkh` is present but `Hkb` levels are low. `Cad` is on by default but repressed by strong `D`, strong `Opa`, strong `Tll` or strong `Hkb`. `D` can be repressed by strong `Opa` or any amount of `Hkb` or `Tll`, and `Cad` must be present for `D` to be expressed strongly. Finally, `Opa` can only be repressed by `Tll` or strong `Hkb`, but it must transit through weak expression before it reaches high levels. This last condition represents the observation that `Opa` protein is synthesised relatively slowly (*Figure 2—figure supplement 4*; *Clark and Akam, 2016*; *Soluri et al., 2020*).

Each simulation begins at *t*0 with `Cad` ubiquitously expressed, and then proceeds through 3 iterations (*t*1–*t*3) in which the expression of `Fkh`, `Wg`, `Cad`, `D`, and `Opa` is synchronously updated based on the current state of the region. *t*0 represents stage 4, *t*1 represents early stage 5, *t*2 represents mid stage 5, and *t*3 represents stage 6. Over the course of a simulation, expression dynamics within each region are shaped both by the (potentially dynamic) inputs from `Tll` and `Hkb`, and by cross-regulation between the other factors. The limited number of expression updates reflects the rapid development of the *Drosophila* blastoderm, which limits the number of regulatory links (i.e., temporally distinct rounds of protein synthesis or decay) within any particular dynamical causal chain (*Nasiadka and Krause, 1999*). Mutant genotypes are simulated by keeping the relevant factor(s) turned off for all timepoints.

### Genotype-by-genotype explanation of simulation output

This section explains the simulated patterning dynamics of each genotype in terms of their underlying regulatory logic. For the wild-type simulation, all expression changes across timepoints *t*1–*t*3 are explained. For the mutant genotypes, only the differences from the wild-type simulation are explained. *Appendix 4—table 1* lists the key features of the simulated expression patterns in each genotype, and, for each prediction, provides figure cross-references to real embryo data showing the same thing.

Wild-type (*Figure 8C*): At *t*1, all three timer genes have begun to be expressed, but they are differentially repressed by the terminal gap genes; `D` and `Opa` are more sensitive to `Tll` and so are repressed everywhere but region 1, while `Cad` is only repressed in region 4, due to the strong `Hkb` expression there. `Fkh` has been activated in regions 3 and 4 due to strong combined `Hkb` and `Tll` expression, together with activation from `Cad`.

At *t2*, the `Cad` expression domain has refined from both the anterior and the posterior. In region 1 it has been repressed by `D`, and in region 3 it has been repressed by the strengthening of `Tll` expression. `Wg` has been activated by `Fkh` in region 3, but remains repressed in region 4 by strong `Hkb` expression.

At *t3*, `D` has been repressed in region 1 by the strong `Opa` expression that has built up over time. Finally, `D` and `Opa` have been de-repressed in region 2, due to the previous clearance of `Tll`.

*fkh*⁻ (**Figure 8D**): Due to the absence of `Fkh`, `Wg` is never activated in region 3.

*cad*^{m-z} (**Figure 8E**): Due to the absence of `Cad`, `Fkh` is never activated in regions 3–4, and `Wg` in turn is never activated in region 3. `D` is also expressed less strongly, both in region 1 and in region 2.

*D*⁻ (**Figure 8F**): Due to the absence of `D`, `Cad` expression persists longer in region 1, although it is later repressed by `Opa`.

*opa*⁻ (**Figure 8G**): Due to the absence of `Opa`, `D` is not repressed completely in region 1. The residual `D` expression in region 1 is weaker than in region 2, because only region 2 receives activation from `Cad`.

*tor*⁻ (**Figure 8H**, modelled as a *hkb*⁻ *tll*⁻ double mutant): In the absence of `Tll` and `Hkb` input, all regions behave exactly like region 1.

*hkb*⁻ (**Figure 8I**): Due to the absence of `Hkb`, `Cad` expression persists for longer in region 4 and `Wg` is de-repressed. There is also a delay in `Fkh` and (therefore) `Wg` expression, which does not affect the final expression pattern.

*tll*⁻ (**Figure 8J**): Due to the absence of `Tll`, the expression of all three timer genes is posteriorly expanded and the size of the `Fkh` domain is reduced. An assumption of graded early `Hkb` expression that represses `D` more anteriorly than `Cad` and `Opa` is necessary to explain the transient `Cad` expression in region 3: `Cad` is first repressed by `D` in regions 1 and 2, and only later by `Opa` in region 3. Because `Fkh` is not expressed outside the `Hkb` domain, `Wg` is never expressed.

*tll*⁻ *opa*⁻ (**Figure 8K**): Patterning resembles the *tll*⁻ mutant through *t2*, but diverges at *t3* due to the absence of `Opa`. Specifically, `Cad` expression in region 3 is allowed to persist, and `D` expression is de-repressed in region 3 after the clearance of `Hkb`. Weak `D` expression also persists in regions 1 and 2, similar to region 1 in *opa*⁻ mutants.

## Modified model for sequential segmentation

In **Figure 8—figure supplement 1**, the AP axis of a sequentially segmenting species is modelled as a growing array of 'cells' with a `Wg` signalling centre at the posterior end, as in **Clark, 2021**. The domain starts at one cell long at *t0*, then adds a cell each iteration by duplicating the most posterior cell. The range of effective `Wg` signalling is finite (in this case, eight cells from the posterior signalling centre), so the zone of `Wg` signalling moves posteriorly with time. Each cell may express `Cad`, `D`, and `Opa`, which are Boolean variables with the following regulatory logic:

    Cad = 0 if (Opa > 0) or ((D > 0) and (Wg < 1)); else Cad = 1.
    D = 0 if (Opa > 0); else D = 1.
    Opa = 0 if (Cad > 0); else Opa = 1.

Thus, `Cad` is repressed by `Opa` and `D` but can be coexpressed with `D` in the presence of `Wg` signalling, `D` is repressed by `Opa`, and `Opa` is repressed by `Cad`. At each iteration, expression in each cell is updated synchronously, based on the current state of the cell.

**Appendix 4—table 1.** Cross-references for simulation output and corresponding expression data.

For each simulated genotype, the 'prediction/observation' column lists noteworthy behaviours of the system that were both predicted by the model and observed in real embryos. The relevant simulation timepoint(s) are listed, along with figure references for the corresponding expression data, and the stages of the embryos/expression traces shown. wt = wild-type.

| Genotype | t | Prediction/observation | Corresponding data | Stage(s) |
|---|---|---|---|---|
| wt | t0 | *cad* expressed ubiquitously. | *Figure 2*; *Figure 2—figure supplement 2*; *Figure 2—figure supplement 4*; *Figure 5—figure supplement 3* | 4.4 |
| wt | t0 | Nested domains of *tll* and *hkb* already established. | *Figure 5—figure supplement 1* | 4.4 |
| wt | t0 | Expression of other factors either absent or just beginning. | *Figure 2* | 4.4 |
| wt | t1 | *fkh* expressed where *tll* (and *hkb*) expression is strong. | *Figure 7A and B*; *Figure 7—figure supplement 1* | 5.4 |
| wt | t1 | *cad* clearing from posterior pole. | *Figure 2*; *Figure 2—figure supplement 4*; *Figure 5—figure supplement 3* | 5.2 |
| wt | t1 | *D* and *opa* expressed in trunk, complementary to *tll*. | *Figure 5A* | 5.2 |
| wt | t1 | Posterior *wg* not yet established. | *Figure 5B* | 5.2 |
| wt | t2 | *tll* domain retracts/narrows over time. | *Figure 5*; *Figure 5—figure supplement 3*; *Figure 5—figure supplement 4*; *Figure 5—figure supplement 5* | 5.2–5.5 |
| wt | t2 | *wg* expressed in *tll*-positive, *hkb*-negative territory. | *Figure 5B* | 5.4 |
| wt | t2 | *cad* clears from trunk and *tll* domain, expressed in between *D*/*opa* to the anterior and *tll*/*wg* to the posterior. | *Figure 2*; *Figure 3A*; *Figure 5* | 5.4–5.5 |
| wt | t2 | A gap opens up between the *tll* and *D*/*opa* domains. | *Figure 5A* | 5.2–5.5 |
| wt | t2 | A gap opens up between the *hkb* and *cad* domains. | *Figure 5B* | 5.3–5.5 |
| wt | t2 | *opa*/Opa expression builds up in the trunk over time. | *Figure 2*; *Figure 3*; *Figure 2—figure supplement 4*; *Figure 5—figure supplement 3*; *Figure 5—figure supplement 3* | 5.1–5.6 |
| wt | t3 | *D* expression clears from the trunk. | *Figure 2*; *Figure 2—figure supplement 3*; *Figure 2—figure supplement 4* | 6 |
| wt | t3 | *D* expression appears in the tail, coexpressed with *cad*. | *Figure 2*; *Figure 2—figure supplement 3*; *Figure 2—figure supplement 4* | 5.5–6 |
| wt | t3 | New *opa* expression appears within the tail, overlapping *cad*. | *Figure 3B* | 5.5–6 |
| *fkh*⁻ | t2–t3 | Posterior *wg* domain absent, patterning otherwise normal. | *Figure 7C and D* | 5.5 |
| *cad*ᵐ⁻ᶻ⁻ | t1 | Early *D* expression is weaker than wt. | *Figure 4C* | 5.2 |
| *cad*ᵐ⁻ᶻ⁻ | t1–t3 | *fkh* expression severely reduced. | *Figure 7A and B* | 5.4 |
| *cad*ᵐ⁻ᶻ⁻ | t2–t3 | Posterior *wg* domain absent. | *Figure 4A and B*; *Figure 7A and B*; *Figure 4—figure supplement 3* | 5.2–6 |
| *cad*ᵐ⁻ᶻ⁻ | t3 | Tail *D* expression reduced compared to wt. | *Figure 4*; *Figure 4—figure supplement 3* | 5.4–6 |
| *cad*ᵐ⁻ᶻ⁻ | t0–t3 | *tll*, *hkb*, and *opa* expression normal. | *Figure 4*; *Figure 7A and B*; *Figure 4—figure supplement 2*; *Figure 7—figure supplement 1* | 5.4–5.5 |
| *D*⁻ | t2 | *cad* expression persists longer in the trunk. | *Figure 4A and B* | 5.5 |
| *opa*⁻ | t3 | Weak *D* expression persists longer in the trunk. | *Figure 4A*; *Figure 4—figure supplement 5* | 5.5–6 |
| *tor*⁻ | t1–t3 | Expression of *cad*, *D*, and *opa* extends to the posterior pole. | *Figure 6B and G* | 5.2–5.5 |
| *tor*⁻ | t2 | *cad* expression clears from the embryo at the normal time. | *Figure 6B* | 5.4 |
| *tor*⁻ | t3 | *D* expression clears from the embryo at the normal time. | *Figure 6B* | 5.5 |
| *tor*⁻ | t2–t3 | The posterior *wg* domain is absent. | *Figure 6B* | 5.2–5.5 |
| *hkb*⁻ | t2–t3 | The posterior *wg* domain extends to the posterior pole. | *Figure 6E and G*; *Figure 7A and B* | 5.4–5.5 |
| *hkb*⁻ | t1 | *cad* expression persists longer in the posterior of the embryo. | *Figure 6E and G*; *Figure 7A and B* | 5.4 |
| *tll*⁻ | t1–t3 | The size of the *fkh* domain is reduced. | *Figure 7A and B* | 5.4 |
| *tll*⁻ | t1–t3 | *cad*, *D* and *opa* expression is posteriorly expanded. | *Figure 6C, D and G*; *Figure 4—figure supplement 5* | 5.3–6 |
| *tll*⁻ | t1–t3 | *cad* and *opa* share a posterior border, *D* is slightly more anterior. | *Figure 6C and D*; *Figure 6—figure supplement 1* | 5.3–5.5 |
| *tll*⁻ | t2 | A transient *cad* stripe is expressed anterior to the *hkb* domain. | *Figure 6C and D*; *Figure 4—figure supplement 5* | 5.3–6 |
| *tll*⁻ | t3 | The *cad* stripe is repressed and there is no posterior *D* domain. | *Figure 6C*; *Figure 4—figure supplement 5* | 6 |
| *tll*⁻ *opa*⁻ | t3 | There is persistent posterior *cad* expression and a posterior *D* domain, unlike in *tll*⁻. | *Figure 4—figure supplement 5* | 6 |

