## [Editor Report]

Through the use of multiplexed in situ hybridisation with careful embryo staging, this article represents exemplary documentation of dynamic gene expression patterns in early fly development. By comparison of these patterns in various mutant combinations, a simple logical model for the specification of expression is proposed. This article will be of broad significance to developmental biologists interested in embryo segmentation and gene regulatory networks underpinning patterning.

---

## [Decision Letter]

**Decision letter after peer review:**

Thank you for submitting your article "A timer gene network is spatially regulated by the terminal system in the *Drosophila* embryo" for consideration by *eLife*. Your article has been reviewed by 3 peer reviewers, and the evaluation has been overseen by a Reviewing Editor and Claude Desplan as the Senior Editor. The following individual involved in the review of your submission has agreed to reveal their identity: Shelby A Blythe (Reviewer #1).

Essential revisions:

1) Please clarify the presentation of Figure 8. Two reviewers found it hard to follow (see detailed comments below).

2) Please move the "by gene" quantitative comparisons into the main figure, perhaps at the expense of the images of embryos. This change could be limited to the figures that compare genotypes, and all images can still be included as supplements.

3) The observation about the size of Caudal germline clones should be removed or reinterpreted. The observation is quite tangential to the point of the paper and likely unrelated to anything having to do with Caudal.

4) The manuscript should better highlight the broader implications. The data are really nice but the way it is currently written (particularly the Results section) somehow downplays the advance to the extent that it sounds more incremental and niche than it actually is.

*Reviewer #1 (Recommendations for the authors):*

This work is, as always, done to the highest degree of rigor and scholarship, and will be invaluable for the field both in terms of its direct focus (terminal segment patterning), as well as a general reference for gene expression patterns of a broad set of patterning genes including some, like fkh, that have overall received less attention over the years.

I am concerned that the paper, in its current form, is overly comprehensive in its presentation of details that it will be inaccessible to all but the most seasoned experts in the field of fly embryo patterning. While I appreciate the comprehensive treatment of each experimental result, I often lost the thread of the writing and had to search for summary paragraphs to make sure that my reading was in line with what the authors meant to convey. One example of this is in the nuanced presentation of the minimal computational model at the end of the Results section. I found myself wanting more of a summary or narrative to this section, rather than a point-by-point description of the model's output. Similar points could be raised about most of the other figures. Because this is a stylistic point, I understand if the authors disagree, but hopefully, some effort at streamlining the Results section could be attempted.

Whether or not such streamlining of the text occurs, I did have some suggestions for improving the presentation of the figures. I found it difficult to follow in the main figures how the plots of gene expression patterns changed from timepoint to timepoint because the plots were always "by embryo" and not "by gene". For instance, in Figures 4 – 6. The critical comparison that we need to make in these two figures is 'by gene over time, between genotypes'. I see that the appropriate 'by gene' plots are provided in the supplemental data. While it will be ok for those reading the paper *online* to be able to flip between all these supplements, those who read a.pdf version in whatever form will be challenged to find this data (or reviewers, for that matter). In this case, it might be better to swap some of the beautiful embryo images for the 'by gene' plots in the supplement in order to substantiate the important points in the main text and move the nice images to the supplement.

If such a drastic change is not wanted, I would at the very least request that in figure 6, the 'matrix' of embryos have equivalent stages across rows between genotypes. For instance row 2 of panels A and B are slightly different stages, but my tendency was to assume we are to compare across these rows.

*Reviewer #2 (Recommendations for the authors):*

1) The shorter, wider phenotype in the germline clone embryos that is attributed to the loss of Cad is something that is usually observed with germline clone embryos generated by the FLP/DFS system. I do not think it is specifically related to the cad mutation. If the data in Fig4 supplement 6 are to remain in the manuscript, the authors should include an analysis of germline clone embryos carrying an unrelated mutation, as I suspect they will have the same shape change.

2) The ectopic activation of Dichaete in the neuroectoderm in cad mutants, and therefore inhibition of neuroectodermal activation of D by Cad is not discussed in relation to the model, which simply shows Cad activating D. Can the authors justify this? What changes in neuroectodermal expression are predicted by the model if Cad has a role in suppressing D activation?

3) If I understand the model correctly, inputs such as Tll can be strong, weak, or off. As the authors mention in the Discussion that there is a phenotypic series of tll alleles, could some of these be used to test whether the model can capture the changes in the network expression patterns when the Tll input is weak instead of strong?

4) In Fig 8 it would help the reader to include, for some key mutants and time points, the same visual representation of the expression patterns based on the observed staining patterns. This would help the reader make a quick and simple comparison - currently, the authors direct the reader to the Source Data 1 but the data are provided in table form.

---

## [Author Response]

Essential revisions:1) Please clarify the presentation of Figure 8. Two reviewers found it hard to follow (see detailed comments below).

We were asked to clarify the presentation of the modelling results from Figure 8. We have moved the genotype-by-genotype description of the model output from the Results section to a new Appendix, and replaced it with a new, narrative summary of the key findings.

2) Please move the "by gene" quantitative comparisons into the main figure, perhaps at the expense of the images of embryos. This change could be limited to the figures that compare genotypes, and all images can still be included as supplements.

We were asked to move the "by gene" quantitative comparisons from the supplementary figures into the main figures that compare gene expression across genotypes (i.e., Figures 4, 6 and 7). We have incorporated the relevant plots as suggested and agree that it is useful to have this information presented in the main figures.

3) The observation about the size of Caudal germline clones should be removed or reinterpreted. The observation is quite tangential to the point of the paper and likely unrelated to anything having to do with Caudal.

We were asked to remove or reinterpret the observation about the size of the *caudal* germline clones. We have rewritten the relevant paragraph to make clear that we do not know whether the morphological changes are related to the loss of Cad expression or the use of the FLP-DFS system. While we agree the finding is of little relevance to the rest of our study, we have retained the observation in the manuscript as we think it is useful information for anyone exploring our embryo dataset or indeed using the FLP-DFS system themselves.

4) The manuscript should better highlight the broader implications. The data are really nice but the way it is currently written (particularly the Results section) somehow downplays the advance to the extent that it sounds more incremental and niche than it actually is.

We were asked to revise the text (especially the Results section) to better highlight the broader implications of our work. We have completely rewritten the Abstract, partially rewritten the Introduction, and completely rewritten the modelling results subsection of the Results (as mentioned in point 1). We have also streamlined the experimental results subsections by editing the text, adding sub-subheadings to signpost the content of each subsection, and more strongly highlighting the “In summary” paragraphs at the end of each subsection. We should stress that the paper is intentionally written so that a casual reader can get by with only the summary paragraphs. However, the rest of the material is necessary if readers want to be walked through the expression data in the main figures, so we prefer to retain it in the main text.

Reviewer #2 (Recommendations for the authors):2) The ectopic activation of Dichaete in the neuroectoderm in cad mutants, and therefore inhibition of neuroectodermal activation of D by Cad is not discussed in relation to the model, which simply shows Cad activating D. Can the authors justify this? What changes in neuroectodermal expression are predicted by the model if Cad has a role in suppressing D activation?

We have now clarified in the Results text that we do not attempt to model the *D* neuroectodermal domain. The neuroectodermal domain is activated at the end of our period of interest and its regulation presumably depends on factors not included in our network, such as dorsoventral patterning genes; it is therefore beyond the scope of our simple model. However, in principle, we see no difficulty with Cad both activating *D*’s early blastoderm expression and repressing *D*’s later neuroectodermal expression, for example through different enhancers and/or indirect effects.

3) If I understand the model correctly, inputs such as Tll can be strong, weak, or off. As the authors mention in the Discussion that there is a phenotypic series of tll alleles, could some of these be used to test whether the model can capture the changes in the network expression patterns when the Tll input is weak instead of strong?

It would be very interesting in future to work to compare gene expression across these different alleles, and also *tll* dosage mutants, to further interrogate the quantitative effects of Tll. (Indeed, while analysing *tll-* homozygotes for this study we noticed that tail patterning was shifted posteriorly in *tll-* heterozygotes compared to wild-type embryos, consistent with Tll having concentration dependent effects.) However, it would be preferable to analyse quantitative effects such as these in the context of a fully quantitative model, rather than the minimalist qualitative model we have used for this study. For the avoidance of doubt, we should also stress that by modelling Tll as a qualitative variable that can be strong, weak, or off, we are not claiming that Tll literally switches between discrete “strong” and “weak” regulatory effects at a specific concentration threshold, but rather that we need at least 3 distinct levels of Tll in our model to reproduce the qualitative patterning phenomena we wish to explain.

4) In Fig 8 it would help the reader to include, for some key mutants and time points, the same visual representation of the expression patterns based on the observed staining patterns. This would help the reader make a quick and simple comparison - currently, the authors direct the reader to the Source Data 1 but the data are provided in table form.

We had considered this option when originally deciding how to present Figure 8, but we decided against it for two reasons. Firstly, rigorously converting real embryo data to the minimal “4 region” representation we used in our model would have required so many (potentially arbitrary) decisions about AP axis positions and intensity thresholds that it felt more straightforward and transparent to simply provide the cross-references to the real data in table form. In this table (now Appendix 4—table 1), we describe noteworthy qualitative aspects of the simulations, and provide figure panel references to the corresponding expression data within the paper. Secondly and more importantly, the qualitative recapitulation of real patterning dynamics by the simulations was actually so accurate that there is only really one discrepancy between the real and simulated data (a spurious delay in the activation of Fkh in the simulated *hkb*- mutant); as a consequence, a visual comparison would have all but duplicated the existing figure. We hope that the rewritten section of the Results now describes the performance of the model much more clearly, obviating the need for a visual comparison.